# Development of a low-cost culture medium from industrial and environmental by-products for sustainable cultivation of Lactic Acid Bacteria

Michele Letitia Tchabou Tientcheu[1], Pierre Marie Kaktcham [1*],
Edith Marius Foko Kouam[2], Laverdure Tchamani Piame [1],
Lysette Chabrone Djodjeu Kamega[1], Aarzoo[3], Agnihotri Shekhar [3],
Singh Bhim Pratap[3], François Zambou Ngoufack[1,2]

**1** Research Unit of Biochemistry, Food Science and Nutrition (URBPMAN), Department of Biochemistry, Faculty of Science, University of Dschang, Dschang, Cameroon, **2** Department of Physiological Sciences and Biochemistry, Faculty of Medicine and Pharmaceutical Sciences, University of Dschang, Dschang, Cameroon, **3** Department of Agriculture and Environmental Sciences (AES), National Institute of Food Technology, Entrepreneurship and Management (NIFTEM), Sonepat, Haryana, India

\* kapima79@yahoo.fr, pierre.kaktcham@univ-dschang.org

## Abstract

Conventional culture media such as de Man, Rogosa and Sharpe are essential for the growth of lactic acid bacteria and the production of metabolites used in the food and pharmaceutical industries. However, their high cost limits their application, particularly in low and middle-income countries. This study aimed to develop a cost-effective and efficient culture medium based on agro-industrial and environmental by-products, pineapple peels, sugarcane molasses, and black soldier fly larvae cake. After physico-chemical analysis of the by-products, excluding sugarcane molasses, a statistical mixture design was used to determine the optimal proportions for supporting lactic acid bacteria growth and bacteriocin production. Growth and metabolite production were monitored via plate count and agar well diffusion assays respectively. The strains tested belonged to the genera *Lactobacillus*, *Bifidobacterium*, *Streptococcus*, *Lactococcus,* and *Bacillus as* an out taxa group. The larvae cake showed high protein (48.47±0.14%) and amino acid (17 types) content, while pineapple peels and molasses were rich in carbohydrates (89.52±0.16% and 86.86±0.07%). Based on regression models, the compromise formulation was defined as 55.15% larvae cake hydrolysate, 19.85% pineapple peel hydrolysate, and 25.00% sugarcane molasses. This medium highly supported lactic acid bacteria growth (9.43–9.86 log CFU/mL) compared to MRS and M17 (9.20–9.69 log CFU/mL), with *Lactobacillus* strains performing better. It also supported bacteriocin activity (11.0–14.5 mm inhibition zones), higher or similar to MRS and M17, with *Lactococcus lactis* subsp. *lactis* MA2 exhibiting the strongest effect. These results highlight the potential of this formulation as a sustainable, low-cost alternative for microbiology and biotechnology, particularly in resource-limited settings. The determination of its

**Data availability statement:** All relevant data are within the paper and its Supporting Information files.

**Funding:** The author(s) received no specific funding for this work.

**Competing interests:** The authors have declared that no competing interests exist.

formula will allow its manufacture once the proximate compositions of the ingredients are known, regardless their origin. Future investigations will focus on optimisation of culture conditions, powder form formulation, and cost evaluation.

## Introduction

In an industrial context undergoing a transition towards more sustainable practices, beneficial microorganisms are attracting increasing interest due to their versatility and safety [1] Among them, lactic acid bacteria (LAB) play a pivotal role in the agri-food, pharmaceutical, cosmetic and animal production sectors, owing to their technological, functional (probiotic) and antimicrobial properties [2]. Beyond their industrial potential, LAB are widely studied in laboratory settings. Their use in lab-scale studies or for biomass and beneficial metabolites productions (for manufacturing starter cultures and probiotic products) at industrial level requires nutrient-rich and complete culture media that meet their needs in terms of carbon and nitrogen sources as well as growth factors. In this respect, they are typically cultivated on conventional media such as MRS [3] and M17 which contains animal peptones, yeast extract, and meat extract. However, these media are expensive and thus represent a major limitation for large-scale production or use in laboratories of resource-limited settings [4]. Indeed, several studies have shown that the high production cost of LAB and their beneficial metabolites is directly related to the nitrogen source, with protein and yeast extract being the most expensive components [5]. In response to these challenges, several studies have explored the use of alternative substrates derived from agro-industrial and environmental wastes/by-products. Culture media have been formulated using materials such as whey [6] organic wastes [7] sheep hooves [8,9], and whey permeate [10] all demonstrating varying degrees of potential. The use of these substrates aligns with sustainable development goals of valorising low-cost residual materials while reducing the environmental impact of microbiological processes. But, these formulations often require supplementation with conventional growth factors (vitamins, minerals) or nitrogen sources (yeast extract, protein hydrolysates) that are still expensive [11], prompting the quest for cheaper substrates which alone or in combination can meet the carbon, nitrogen and growth factor needs and support the growth and metabolites production by LAB. Kaktcham et al. [12] reported that media formulated solely from fish waste and sugarcane molasses could efficiently support bacteriocin production by *Lactococcus lactis* subsp. *lactis* 2MT. Similarly, Valle Vargas et al. [13] demonstrated that media formulated from whey, sugarcane molasses, and palm kernel cake could support the growth of fish probiotics *Lactococcus lactis* A12, *Priestia megaterium* M4, and *Priestia* sp. M10, while Acosta-Piantini et al. [14] showed that hydrolysed sugarcane molasses could serve as an economical substrate for mass production of *Lacticaseibacillus paracasei*. However, such studies are still scarce not only, but also the combination of multiple unexplored wastes/by-products that are released in our environment and contribute to pollution can also be valorised in the formulation of low-cost culture media. Such an approach could

provide a balanced source of carbon, nitrogen, and growth factors, thereby enhancing both LAB growth and metabolite production. Three local agro-industrial and environmental by-products that have not yet been used in the formulation of culture media alone or in combination were selected in the present study. The black soldier fly larvae *(Hermetia illucens)* cake, a protein-rich residue from the biofuel and animal feed industries, represents an emerging biotechnological resource in Africa. It contains high-quality proteins, lipids, minerals (Fe, Zn, Mg), and bioactive compounds which can stimulate microbial growth and metabolism [15]. Pineapple peels, abundantly generated by fruit juice industries and local markets, are rich in fermentable sugars (glucose, fructose, sucrose), organic acids, and B-group vitamins, as well as phenolic compounds that may enhance bacterial stress resistance [16]. Sugarcane molasses, a by-product of sugar refineries such as SOSUCAM in Cameroon, provides readily assimilable carbohydrates, minerals (Ca, K, Mg), and trace elements essential for enzymatic functions in LAB [17]. Their combination may allow the development of economical and sustainable culture media in line with the principles of the circular bioeconomy [18].

This study was therefore aimed to formulate from agro-industrial and environmental by-products a low-cost culture medium able to sustain the growth and beneficial metabolites production of a wide genera of lactic acid bacteria.

## Materials and methods

### Bacterial strains, media, and growth conditions

Ten bacterial strains with technological and probiotic interests were used in this study (Table 1), including nine lactic acid bacteria belonging to genera *Lactobacillus*, *Lacticaseibacillus*, *Lactiplantibacillus*, *Streptococcus*, *Lactococcus* and *Bifidobacterium*. *Bacillus subtilis* was also used and considered as an out taxa group.

From frozen stocks, the LAB strains were revived by two successive subcultures of 18 h each [19] in MRS broth (HIMEDIA, India) for *lactobacilli* and in M17 broth (HIMEDIA, India) for *lactococci, pediococci and streptococci*, before being used for subsequent experiments. The pathogenic bacteria were revived as stated above but in BHI broth (TITAN BIO-TECH LTD, India).

**Table 1. Bacterial strains, sources, applications and growth conditions.**

| Strains | Sources | Status/Application | Growth Temperature (°C)/ Anaerobiosis |
|---|---|---|---|
| *Lactiplantibacillus plantarum* **5602** | NCIM, Pune | Probiotic, technological, bacteriocin producer | 37°C |
| *Lacticaseibacillus rhamnosus* **347** | NCDC | Probiotic, Bacteriocin producer | 37°C |
| *Lactobacillus acidophilus* **291** | NCIM, Pune | Probiotic | 37°C |
| *Lactabacillus gasseri* **5359** | NCIM, Pune | Probiotic | 37°C |
| *Lactobacillus delbrueckii* subsp. *bulgaricus* **293** | NCDC | Technological | 40°C |
| *Streptococcus thermophilus* **295** | NCDC | Technological | 40°C |
| *Lactococcus lactis* subsp. *lactis* **MA2** | URBPMAN | Technological Bacteriocin producer | 30°C |
| *Lactococcus lactis* subsp. *lactis* **MF5** | URBPMAN | Technological Bacteriocin producer | 30°C |
| *Bifidobacterium bifidum* **231** | NCDC | Probiotic, Bacteriocin producer | 37°C/ Anaerobic |
| *Bacillus subtilis* **215** | NCDC | Technological | 37°C |
| *Pediococcus acidilactici* | URBPMAN | Sensitive strain | 30°C |
| *Escherichia coli* **ATCC 11775** | ATCC | Pathogenic | 37°C |
| *Staphylococcus aureus* **NCDC 109** | NCDC | Pathogenic | 37°C |

**ATCC:** American Type Culture Collection. **NCDC:** National Collection of Dairy Cultures of the National Dairy Research Institute, India. **NCIM:** National Collection of Industrial Microorganisms Pune, India, **URBPMAN:** Culture collection of the Research Unit of Biochemistry, Food Science and Nutrition of the University of Dschang, Cameroon.

## Sampling, physico-chemical analysis, and preparation of hydrolysates of by-products

Pineapple peels were collected from vendors at Market B in the city of Dschang (Menoua Division). Sugarcane molasses was obtained from the SOSUCAM sugar company in Mbanjock, located in the Centre Region of Cameroon, stored at 4°C in tightly sealed glass bottles to prevent microbial contamination and moisture absorption. Before use, the solution was diluted to 4% (w/v) with distilled water, thoroughly mixed, and filtered through Whatman paper No.1. Black soldier fly larvae cake (BC) was sourced from biofuel producers at the Faculty of Agronomy and Agricultural Sciences (FAAS). University of Dschang (West Region of Cameroon), stored in hermetically sealed plastic bags inside a desiccator to maintain dryness until use. All samples were transported to the Research Unit for Biochemistry, Food Science and Nutrition (URB-PMAN), where they were processed. Pineapple peels were oven-dried at 45°C for 48 h, ground using a blender (Silver Crest, China), and sieved through a 200-micron mesh. The resulting powder was stored in a desiccator until use.

### Physico-chemical analysis of the by-products

The physico-chemical analysis of pineapple peels, BC and sugarcane molasses was conducted according to IUPAC methods [20] for fats and sugars, AOAC methods [21] for proteins and ash. The Minerals (Ca, P, K, Na, Zn, Mg, Mn and Fe) were quantified using inductively coupled plasma optical emission spectrometry (Optima 8000 ICP-OES, Perkin Elmer, USA) as described by Savic et al. [22]. Sugar profiles were determined by high-performance liquid chromatography (HPLC) following the protocols of Reuter et al. [23]. Briefly, after filtration through a 0.45 μm membrane filter, liquid samples were directly injected into a high-performance liquid chromatography (HPLC) system; the analysis was performed using an Aminex HPX-87P column (300 × 7.8 mm) (Bio-Rad, USA), maintained at 80°C, with ultrapure water as the mobile phase in isocratic mode at a flow rate of 0.6 mL/min. Detection was carried out using a refractive index detector (RID). Each sample was injected at a volume of 20 μL. Sugar identification was based on retention time comparison with authentic standards (glucose, fructose, sucrose, lactose, maltose), and quantification was achieved using external calibration curves prepared with standard concentrations ranging from 0.1 to 10 g/L. Results were expressed in g/100 g of sample.

The amino acid composition was determined according to the method described by Savych et al. [24] with slight modifications. Samples were first hydrolysed using hydrochloric acid (HCl, 6N) at 110°C for 24 h under a nitrogen atmosphere to release bound amino acids. After hydrolysis, the samples were cooled, filtered, evaporated to dryness under vacuum, and reconstituted in a borate buffer. Amino acids were then derivatised using o-phthalaldehyde (OPA) in combination with 2-mercaptoethanol, enabling fluorescence detection. The analysis was carried out using high-performance liquid chromatography (HPLC) on a C18 reverse-phase column, employing a gradient elution with a mobile phase consisting of phosphate buffer and an organic solvent mixture (acetonitrile/methanol/water). Fluorescence detection was performed at 340 nm excitation and 455 nm emission. Amino acids were identified by comparing retention times with commercial standards, and quantification was performed using external calibration. Results were expressed as mg of amino acid per 100 g of dry sample.

### Preparation of pineapple peels hydrolysate

The method described by Pumiput et al. [25] was used with slight modifications to prepare pineapple peel hydrolysate. A 4 g portion of the substrate was steam-treated in an autoclave (Sonoclav, Germany) at 121°C (15 psi) for 20 min. After cooling, distilled water was added to adjust the final volume to 100 mL. The mixture was then boiled at 100°C for 30 min in a water bath (HH-W420 Coslab, India). The resulting boiled mixture was sieved through cotton cloth and acid hydrolysis was thereafter performed by autoclaving the filtrate at 121°C (15 psi) for 30 min with HCl (1% v/v of 6N solution). The pH of the hydrolysate was adjusted to 6.5 using calcium carbonate (98%), followed by filtration through Whatman No. 1 filter paper.

## Preparation of black soldier fly larvae cake hydrolysate

Acid hydrolysis of the BC was performed based on conditions selected from the literature and preliminary laboratory work (clearness of the solution and amino acids profile of hydrolysate). Thus, 8 g of BC was hydrolysed in 99 mL of distilled water with 1 mL HCl (6N) at 160.5°C for 17.5 min in an oven (Memmert, Germany). After hydrolysis, the samples were centrifuged, and the supernatants were neutralised using NaOH (6N).

## Development of the low-cost culture medium

**Mixing design.** In search of the optimal proportions of ingredients that can support the growth and metabolite production by the LAB strains, a simplex centroid design was implemented. Specifically, a three-factor extreme peak mixing plan (BC hydrolysate, pineapple peel hydrolysate and 4% sugarcane molasses solution) was used to define the proportions of each mixture component [26]. Preliminary work and the literature were used to establish the extremes. Table 2 below shows the experimental range of factors.

Once the ranges of the different factors were determined, the calculations for converting the coded variables into real variables for a 3-factor mixing design were carried out, and the results are reported in Table 3 below, with nine experiments carried out. All these experiments were repeated three times.

## Assessment of the performances of media formulated on Biomass production

Nine media were formulated according to the experimental matrix and inoculated with 1% (v/v) of a standardised (McFarland 4) suspension of each strain and incubated at 30°C or 37°C for 18 h, in aerobic or anaerobic conditions, depending on the strain. Viable cell counts were determined using the plate count method in MRS and M17 agar at 30 or 37°C for 48 h. The bacterial growth was estimated by difference between the counts at final and initial times of incubation and expressed in log CFU/mL. MRS and M17 broth media were used as control.

**Table 2. Experimental range of mixing factors.**

| Factors | Levels | | |
| --- | --- | --- | --- |
| | Symbols | Lower | Upper |
| **Black soldier fly larvae cake hydrolysate (% v/v)** | $X_1$ | 50 | 75 |
| **Pineapple peel hydrolysate (% v/v)** | $X_2$ | 12.5 | 25 |
| **4% sugarcane molasses (% v/v)** | $X_3$ | 12.5 | 25 |

**Table 3. Experimental matrix for the mixing design.**

| Runs | Black soldier fly larvae cake hydrolysate (%v/v) | Pineapple peel hydrolysate (%v/v) | Sugarcane molasses (%v/v) |
| --- | --- | --- | --- |
| 1 | 62.500 | 15.625 | 21.875 |
| 2 | 56.250 | 21.875 | 21.875 |
| 3 | 68.750 | 15.625 | 15.625 |
| 4 | 50.000 | 25.000 | 25.000 |
| 5 | 62.500 | 18.750 | 18.750 |
| 6 | 62.500 | 12.500 | 25.000 |
| 7 | 62.500 | 25.000 | 12.500 |
| 8 | 62.500 | 21.875 | 15.625 |
| 9 | 75.000 | 12.500 | 12.500 |

## Assessment of the performances of media formulated on bacteriocin production

The agar well diffusion method described by Kaktcham et al. [27] was used. After 18 h of incubation, cultures were centrifuged at 10.000×g for 10 minutes at 4°C. The supernatants were neutralised to pH 6.5 using NaOH (1N), filtered through a 0.22 µm membrane and heat-treated at 100°C for 5 min (to destroy $H_2O_2$). Wells of 6 mm diameter were punched into BHI or MRS agar plates previously seeded on the soft second layer with an indicator strain (*S. aureus* NCDC 109, *E. coli* ATCC 11775 and *P. acidilactici*) and 100 µL of treated supernatant was added to each well. After 24 h of incubation at 30 or 37°C. Inhibition zones ≥ 2 mm were considered indicative of bacteriocin activity. MRS and M17 broth media were used as controls.

## Model fitting of mixture design and definition of the compromise medium

The experimental data obtained from the simplex centroid mixture design with three components were used and modeled using a commonly applied linear model:

$$Y = _0 + \beta_1 x_1 + \beta_2 x_2 + \beta_3 x_3 + \beta_{12} x_1 x_2 + \beta_{13} x_1 x_3 + \beta_{23} x_2 x_3.$$

The coefficients β1, β2, and β3 represent the linear effects of each component on the response, whereas β12, β13, and β23 indicate the interactions between pairs of components. Positive coefficients suggest stimulation of growth or metabolite production, while negative coefficients indicate inhibition.

Model fitting was performed using the least squares method with the aid of Minitab 18 software. Response surface plots were generated based on the fitted model to visualize the combined effect of the different proportions on the dependent variable. Analysis of variance (ANOVA) was carried out to assess the validity of the model. A significance level of 5% ($p < 0.05$) was used to determine whether the mixture components have a statistically significant effect on the measured response. For the model to be validated, the following criteria should be respected: coefficient of determination ($R^2$) ≥ 0.75; absolute mean deviation analysis of the (AMDA) =0 and the bias factor 0.75<Bf<1.25. Statistical analyses enabled the optimisation of the formulation by targeting specific responses to be maximised or minimised. The software proposed optimal conditions under which the experimental results closely matched the predicted values. The compromise in a mixture design was achieved through a desirability function approach (>0.75), which combines multiple response variables into a single objective function. This method allows a balanced optimisation based on predefined priorities.

## Establishment of the formula of the compromise medium

To ensure the reproducibility of the culture medium formulation using the same categories of agro-industrial by-products regardless of their geographical origin or batch-specific composition the final medium formula was established based on the physico-chemical composition of the tested hydrolysates and by-products (hydrolysate of BC, hydrolysate of pineapple peels and 4% sugarcane molasses solution). The fresh matter equivalent to the volume of each proportion of the ingredients was obtained by multiplying the density by this volume. Then, the dry matter content of the volume of each proportion (X, Y or Z) of the ingredients was obtained by multiplying the % dry matter of the ingredient by the fresh matter equivalent to this volume. Thereafter, the quantity of protein for instance ($a_iX$) in each volume of ingredient was obtained by multiplying the % protein in the dry matter by the dry matter (X) of the ingredient volume. Given that the total volume of the medium is 100 mL, and to report the values as concentrations, the final quantities of proteins, lipids and carbohydrates were multiplied by 10. Hence the formula:

$$10(a1X + a2Y + a3Z) = P \tag{1}$$

$$10(b1X + b2Y + b3Z) = L \qquad (2)$$

$$10(c1X + c2Y + c3Z) = C \qquad (3)$$

Where: $X$ = dry matter content (g) in 55.1508 mL of BC hydrolysate; $Y$ = dry matter content (g) in 19.849 mL of pineapple peel hydrolysate; $Z$ = dry matter content (g) in 25.00 mL of 4% sugarcane molasses solution. $a_i$, $b_i$, $c_i$ represent the protein, lipids, and carbohydrates contents (%DM) in hydrolysates and sugarcane molasses solution; $P$, $L$ and $C$ represent total proteins, lipids, and carbohydrates contents (g/L) of the formulated compromise culture medium.

## Statistical analysis

Statistical analysis was conducted using Minitab 18 and SPSS version 22 softwares. Minitab was used to design the experimental and mixture designs and to analyze response data. Desirability functions were generated to determine optimal formulations for bacterial growth and bacteriocin production. Model performance was evaluated using the coefficient of determination ($R^2$) as previously stated. Elsewhere, the results of bacterial growth and bacteriocin production obtained from the nine runs were expressed as means ± standard deviation. Analysis of variance (ANOVA) was applied to assess significant differences between means at a 5% significance level ($p < 0.05$), followed by Duncan's multiple range test for pairwise comparisons when the means were different, using SPSS (version 22).

## Results

### Physico-chemical analysis of the by-products

Table 4 below and S1 Table present the macronutrient and mineral compositions of the three by-products assayed. BC had the highest protein content (48.47 ± 1.4%) and the lowest carbohydrate content (26.27 ± 1.17%). In contrary, pineapple peels and sugarcane molasses were rich in sugars (89.52 ± 0.16% and 86.86 ± 0.07% respectively) and contained lower protein levels (4.89 ± 0.05% and 7.02 ± 0.01%). These by-products were also found to be sources of minerals, with potassium (K) being the most abundant (2.872 ± 0.05 mg/100g) in sugarcane molasses, followed by calcium (Ca) and magnesium (Mg) in BC (1.252 ± 0.14 mg/100g and 0.437 ± 0.03 mg/100g respectively). Manganese (Mn), an essential trace element for lactic acid bacteria, was also detected in small amounts (0.0171 ± 0.24 mg/100g).

**Table 4. Physico-chemical composition of BC, pineapple peel and sugarcane molasses.**

| Parameters | Black soldier fly larvae cakes | Pineapple peels | Sugarcane molasses |
|---|---|---|---|
| Lipids (% DM) | 16.470 ± 0.190[a] | 0.750 ± 0.150[b] | 0.000 ± 000[c] |
| Proteins (% DM) | 48.470 ± 0.140[a] | 4.890 ± 0.050[c] | 7.020 ± 0.010[b] |
| Carbohydrates (% DM) | 26.270 ± 0.170[c] | 89.520 ± 0.160[a] | 86.860 ± 0.070[b] |
| Na (mg/100g) | 1.252 ± 0.140[a] | 0.547 ± 0.660[c] | 0.764 ± 0.260[b] |
| Mg (mg/100g) | 0.437 ± 0.030[a] | 0.140 ± 0.410[c] | 0.175 ± 0.080[b] |
| K (mg/100g) | 1.184 ± 0.510[c] | 1.429 ± 0.370[b] | 2.872 ± 0.050[a] |
| P (mg/100g) | 0.710 ± 0.210[a] | 0.136 ± 0.570[b] | 0.067 ± 0.320[c] |
| N (mg/100g) | 0.214 ± 0.310[a] | 0.059 ± 0.090[b] | 0.057 ± 0.450[b] |
| Mn (mg/100g) | 0.017 ± 0.240[a] | 0.017 ± 0.030[a] | 0.006 ± 0.120[b] |
| Fe (mg/100g) | 0.067 ± 0.000[b] | 0.101 ± 0.440[a] | 0.055 ± 0.330[c] |
| Zn (mg/100g) | 0.005 ± 0.480[a] | 0.003 ± 0.070[c] | 0.004 ± 0.200[b] |

[a,b,c]: different letters in the same line indicate significant differences ($p < 0.05$).

The amino acid profile of the three by-products (S2 and S3Tables) shows that BC contained all 17 amino acids for which standards were available, with the significantly (p < 0.05) highest concentrations of histidine, glycine, aspartic acid, alanine and leucine. In contrast, sugarcane molasses and pineapple peel contained only 8 and 6 amino acids respectively and displayed significantly (p < 0.05) higher concentrations of lysine (1.99 ± 0.17 and 1.73 ± 0.01 µmol/mL respectively), with the significantly highest (p < 0.05) value recorded in pineapple peel.

The sugar profiles of the by-products are shown in Table 5 and S4 Table. All three substrates contained mainly sucrose, glucose, and fructose, but not maltose and galactose. Molasses had the significantly (p < 0.05) higher sucrose content, followed by pineapple peels. The latter was rather the significantly richest in glucose and fructose, followed by sugarcane molasses. The BC does not contain glucose.

## Performances of the low-cost culture media formulated

**Experimental trials and biomass production.** Based on the experimental matrix provided by the mixture design, nine media were formulated and used to assess strains' biomass and bacteriocin productions as responses. The results of colony count after 48 h of incubation are presented in Table 6 and S5 Table. The tested strains could grow correctly in formulated media and some of them even matched or outperformed their growth on the reference commercial media MRS and M17. For *Lactiplantibacillus plantarum* 5602, the bacterial count ranged from 9.04 ± 0.03 to 9.36 ± 0.10 log CFU/mL. The significantly (p < 0.05) highest value was obtained with trial 5 and was significantly (p < 0.05) higher than the count obtained in MRS (9.20 ± 0.24 log CFU/mL). For *Lacticaseibacillus rhamnosus* 347, the bacterial count ranged from 9.38 ± 0.27 to 9.69 ± 0.28 log CFU/mL. The significantly (p < 0.05) highest value was obtained with trial 9 and was equal to the count obtained in MRS (9.69 ± 0.15 log CFU/mL). Concerning *Lactobacillus gasseri* 5359, the bacterial count range was 9.36 ± 0.21 to 9.68 ± 0.23 log CFU/mL, the significantly (p < 0.05) highest values obtained with trials 2 and 5, also significantly (p < 0.05) higher than the value obtained with MRS (9.59 ± 0.20 log CFU/mL). With reference to *Lactobacillus acidophilus* 291, the growth range was 9.52 ± 0.30 to 9.73 ± 0.40 log CFU/mL, the significantly (p < 0.05) highest value, also significantly (p < 0.05) higher than that of MRS (9.62 ± 0.13 log CFU/mL), obtained with trials 2 and 5. As to *Lactobacillus delbrueckii* subsp. *bulgaricus* 293, the growth range was 9.52 ± 0.12 to 9.67 ± 0.21 log CFU/mL, the significantly (p < 0.05) highest value, also significantly (p < 0.05) higher than that of MRS (9.55 ± 0.25 log CFU/mL), obtained with trials 2 and 5. Considering *Streptococcus thermophilus* 295, the growth ranged 9.33 ± 0.19 to 9.64 ± 0.21 log CFU/mL, with the statistically highest values significantly (p < 0.05) higher than that in M17 (9.31 ± 0.11 log CFU/mL) and obtained with trials 2 and 5. As for *Lactococcus lactis* subsp. *lactis* MA2, the growth ranged 9.31 ± 0.22 to 9.48 ± 0.43 log CFU/mL, with the statistically highest value significantly (p < 0.05) higher than that in M17 (9.12 ± 0.59 log CFU/mL) and obtained with trial 5. With respect to and *Lactococcus lactis* subsp. *lactis* MF5, the growth range was 9.40 ± 0.31 to 9.59 ± 0.21 log CFU/mL, with the statistically highest value significantly (p < 0.05) higher than that in M17 (9.49 ± 0.28 log CFU/mL) and obtained with trial 2. Regarding *Bifidobacterium bifidum* 231, the biomass range was 9.52 ± 0.01 to 9.65 ± 0.17 log CFU/mL, the significantly (p < 0.05) highest values, also higher (p < 0.05) than that in MRS (9.53 ± 0.29 log CFU/mL), obtained

Table 5. Sugar profiles of the various by-products (BC, pineapple peels, sugarcane molasses).

| Sugars | Black soldier fly larvae cake (mg/kg) | Pineapple peel (mg/kg) | Sugarcane molasses (mg/kg) |
|---|---|---|---|
| Fructose | 44.401 ± 0.10[c] | 168.071 ± 0.13[a] | 112.47 ± 0.05[b] |
| Glucose | ND | 253.178 ± 0.20[a] | 195.793 ± 0.07[b] |
| Sucrose | 15.241 ± 0.09[c] | 48.710 ± 0.06[b] | 579.376 ± 0.10[a] |
| Maltose | ND | ND | ND |
| Galactose | ND | ND | ND |

[a-c]: In the same line Different letters indicate significant differences (p < 0.05). ND = Not detected.

Table 6. Strains biomass production in the media formulated based on the experimental matrix.

| Trials | BCH (%v/v) | PPH (%v/v) | SM (%v/v) | Lactiplantibacillus plantarum 5602 | Lacticaseibacillus rhamnosus 347 | Lactobacillus acidophilus 291 | Lactobacillus gasseri 5359 | Bifidobacterium bifidum 231 | Bacillus subtilis 215 | Lactobacillus delbrueckii subsp. bulgaricus 293 | Streptococcus thermophilus 295 | Lactococcus lactis subsp. lactis MA2 | Lactococcus lactis subsp. lactis MF5 |
|---|---|---|---|---|---|---|---|---|---|---|---|---|---|
| | | | | Biomass (log CFU/mL) | | | | | | | | | |
| 1 | 62.500 | 15.625 | 21.875 | 9.35±0.07e | 9.63±0.70e | 9.66±0.20e | 9.6±0.21ed | 9.64±0.32e | 9.58±0.10dc | 9.6±0.13d | 9.57±0.30dc | 9.41±0.28d | 9.55±0.10c |
| 2 | 56.250 | 21.875 | 21.875 | 9.33±0.12e | 9.65±0.60f | 9.72±0.12f | 9.67±0.17g | 9.63±0.12c | 9.6±0.03dc | 9.67±0.21fe | 9.62±0.01e | 9.43±0.48d | 9.59±0.21d |
| 3 | 68.750 | 15.625 | 15.625 | 9.19±0.02c | 9.43±0.40a | 9.71±0.18a | 9.6±0.38ed | 9.53±0.44a | 9.56±0.14cb | 9.6±0.33ba | 9.50±0.42bc | 9.41±0.12d | 9.44±0.33ab |
| 4 | 50.000 | 25.000 | 25.000 | 9.31±0.10e | 9.62±0.33e | 9.67±0.10e | 9.65±0.41f | 9.57±0.06bcd | 9.57±0.17d | 9.63±0.44e | 9.59±0.33d | 9.42±0.57e | 9.52±0.43c |
| 5 | 62.500 | 18.750 | 18.750 | 9.36±0.10f | 9.69±0.28g | 9.73±0.40f | 9.68±0.23g | 9.64±0.28d | 9.63±0.21cb | 9.65±0.21f | 9.64±0.21e | 9.48±0.43d | 9.55±0.22bc |
| 6 | 62.500 | 12.500 | 25.0000 | 9.27±0.05b | 9.51±0.25c | 9.6±0.35c | 9.58±0.41b | 9.65±0.17e | 9.57±0.07cb | 9.63±0.28ed | 9.33±0.19a | 9.42±0.18c | 9.55±0.16 cd |
| 7 | 62.500 | 25.000 | 12.500 | 9.24±0.30b | 9.47±0.40b | 9.57±017b | 9.63±0.25f | 9.58±0.21 cd | 9.56±0.10cb | 9.53±0.47b | 9.54±0.00cb | 9.31±0.22c | 9.51±0.15c |
| 8 | 62.500 | 21.875 | 15.625 | 9.33±0.12e | 9.62±0.13d | 9.65±0.28d | 9.61±0.11e | 9.6±0.40 cd | 9.57±0.00a | 9.59±0.30dc | 9.58±0.04d | 9.34±0.13c | 9.54±0.04a |
| 9 | 75.000 | 12.500 | 12.500 | 9.04±0.03a | 9.38±0.27a | 9.52±0.30a | 9.54±0.32a | 9.52±0.01ab | 9.52±0.04a | 9.52±0.12a | 9.51±0.21b | 8.92±0.17a | 9.4±0.31a |
| MRS/M17 | | | | 9.20±0.24d | 9.69±0.15g | 9.62±0.13c | 9.59±0.20d | 9.53±0.29ab | 9.54±0.00ba | 9.55±0.25cb | 9.31±0.11a | 9.12±0.59b | 9.49±0.28abc |

a-g: In the same column, means with different lower-case letters differ significantly ($p < 0.05$) Culture on MRS medium: Lacticaseibacillus rhamnosus 347, Lactiplantibacillus plantarum 5602, Lactobacillus acidophilus 291, Bifidobacterium bifidum 231, Lactobacillus delbrueckii subsp. bulgaricus 293, Lactobacillus gasseri 5359, Bacillus subtilis 215. Culture on M17 medium: Lactococcus lactis subsp. lactis MA2, Lactococcus lactis subsp. lactis MF5 and Streptococcus thermophilus 295.

**BCH** black soldier fly larvae cake hydrolysate (% v/v); **PPH**: pineapple peel hydrolysate (% v/v); **SM**: 4% sugarcane molasses (%v/v).

with trials 1 and 6. With reference to *Bacillus subtilis* 215, the biomass range was 9.52±0.04 to 9.63±0.21 log CFU/mL, the significantly ($p<0.05$) highest value, also higher ($p<0.05$) than that in MRS (9.54±0.00 log CFU/mL), obtained with trials 5. These results confirm the strong growth-supporting potential of the formulated alternative media, particularly for probiotics and technologically relevant strains. Trial 5 medium proved to highly support the growth of 70% of the strain, compared to 50% with trial 2.

Table 7 presents the growth ranges (CFU/mL), maximum values, and comparison with the control for each lactic acid bacteria strain. Strains showing higher growth than the control included *Lactiplantibacillus plantarum* 5602, *Lactobacillus acidophilus* 291, *Lactobacillus gasseri* 5359, *Lactococcus lactis* subsp. *lactis* MA2, *Bifidobacterium bifidum* 231, *Lactobacillus delbrueckii* subsp. *bulgaricus* 293 and *Streptococcus thermophilus* 295. Strains with growth comparable to the control were *Lacticaseibacillus rhamnosus* 347, *Lactococcus lactis* subsp. *lactis* MF5 and *Bacillus subtilis* 215. These results helped to identify the most favorable substrates for biomass production and guided the formulation of optimised culture media.

## Proposed mathematical models for biomass production and contribution of factors

After analysing the effect of each proportion as well as the interactions of the by-products (proportions of BC hydrolysate, pineapple peel hydrolysate and sugarcane molasses solution) on the response (biomass), the mathematical model for each strain was predicted using the following regression equations:

Biomass of *Lactiplantibacillus plantarum* 5602

$$-0.042x1 - 0.96x2 + 0.033x3 + 0.014x1x2 + 0.015x2x3 = Y1 \tag{4}$$

Biomass of *Lacticaseibacillus rhamnosus* 347

$$-0.098x1 - 2.06x2 - 0.00x3 + 0.032x1x2 + 0.03x2x3 = Y2 \tag{5}$$

Biomass of *Lactobacillus acidophillus* 291

$$-0.069x1 - 01.74x2 + 0.006x3 + 0.02x1x2 + 0.02x2x3 = Y3 \tag{6}$$

**Table 7. Comparative growth performance of Lactic Acid Bacteria in the formulated and control (MRS/M17) media.**

| Strains | Growth range (log CFU/mL) | Max value in formulated media (log CFU/mL) | Value in Control media (log CFU/mL) | p-value | Comparative effect |
|---|---|---|---|---|---|
| *Lactiplantibacillus plantarum* 5602 | 9.04±0.03-9.36±0.10 | 9.36±0.10 | 9.20±0.24 | 0.037 | Superior |
| *Lacticaseibacillus rhamnosus* 347 | 9.38±0.27-9.69±0.28 | 9.69±0.28 | 9.69±0.15 | 0.884 | Comparable |
| *Lactobacillus acidophilus* 291 | 9.52±0.30-9.73±0.40 | 9.73±0.40 | 9.62±0.13 | 0.041 | Superior |
| *Lactobacillus gasseri* 5359 | 9.54±0.32-9.68±0.23 | 9.68±0.23 | 9.59±0.20 | 0.138 | Comparable |
| *Lactobacillus delbrueckii* subsp. *bulgaricus* 293 | 9.52±0.12-9.67±0.21 | 9.67±0.21 | 9.55±0.25 | 0.048 | Superior |
| *Streptococcus thermophilus* 295 | 9.330.19±−9.64±0.21 | 9.64±0.21 | 9.31±0.11 | 0.012 | Superior |
| *Lactococcus lactis* subsp. *lactis* MA2 | 8.92±0.17-9.48±0.43 | 9.48±0.43 | 9.12±0.59 | 0.028 | Superior |
| *Lactococcus lactis* subsp. *lactis* MF5 | 9.40±0.31-9.59±0.21 | 9.59±0.21 | 9.49±0.28 | 0.185 | Comparable |
| *Bifidobacterium bifidum* 231 | 9.52±0.01-9.65±0.17 | 9.65±0.17 | 9.53±0.29 | 0.047 | Superior |
| *Bacillus subtilis* 215 | 9.52±0.04−9.63±0.21 | 9.63±0.21 | 9.54±0.00 | 0.149 | Comparable |

**CFU/mL**: Colony Forming Units per milliliter. $p<0.05$ was considered significant.

Biomass of *Lactabacillus gasseri* 5359

$$0.01x1 - 0.83x2 + 0.004x3 + 0.014x1x2 + 0.0142x2x3 = Y4 \tag{7}$$

Biomass of *Lactobacillus delbrueckii* subsp. *bulgaricus* 293

$$0.35x1 - 0.74x2 + 0.11x3 + 0.012x1x2 + 0.01x2x3 = Y5 \tag{8}$$

Biomass of *Streptococcus thermophilus* 295

$$0.31x1 - 1.80x2 - 0.20x3 + 0.02x1x2 + 0.03x2x3 = Y6 \tag{9}$$

Biomass of *Lactococcus lactis* subsp. *lactis* MA2

$$0.09x1 - 1.30x2 + 0.13x3 + 0.02x1x2 + 0.01x2x3 = Y7 \tag{10}$$

Biomass of *Lactococcus lactis* subsp. *lactis* MF5

$$0.03x1 - 0.41x2 + 0.16x3 + 0.008x1x2 + 0.01x2x3 = Y8 \tag{11}$$

Biomass of *Bifidobacterium bifidum* 231

$$0.35x1 - 0.53x2 + 0.19x3 + 0.01x1x2 + 0.001x2x3 = Y9 \tag{12}$$

Biomass of *Biomass of Bacillus subtilis* 215

$$0.14x1 - 0.69x2 + 0.05x3 + 0.01x1x2 + 0.01x2x3 = Y10 \tag{13}$$

where: $Y$ = Biomass (log CFU/mL); $x1$ = Black soldier fly larvae cake hydrolysate (% v/v); $x2$ = Pineapple peel hydrolysate (% v/v); $x3$ = Sugarcane molasses (% v/v).

The analysis of the regression equations highlights the respective contributions of linear effects and interactions among the media constituents on the cell viability of the tested strains. For all the strains, the SM ($x_3$), the interaction of BCH and PPH ($x_1x_2$), as well as the interaction of PPH and SM ($x_2x_3$) showed positive effects on biomass production, except for *Lacticaseibacillus rhamnosus* 347. In addition, the linear effect of BCH ($x_1$) was also positive on biomass production of *Lactobacillus gasseri* 5359, *Streptococcus thermophilus* 295, *Bifidobacterium bifidum* 231, *Bacillus subtilis* 215, *Lactobacillus delbrueckii* subsp. *bulgaricus 293*, and *Lactococcus lactis* subsp. *lactis* MA2 and MF5. This indicates that BCH and SM stimulated biomass production of the strains, while PPH did not contributed to this enhanced production. In addition to these positive linear effects, the positive interactions ($x_1x_2$ and $x_2x_3$,) observed for these by-products suggests that they act in synergy to enhance bacterial growth.

## Validation of mathematical models for biomass production

The regression models established for predicting biomass production of lactic acid bacteria showed a good fit between experimental and predicted data. Determination coefficients ($R^2$) were all greater than 0.83, while bias factors (Bf) ranged between 0.75 and 1.25, confirming the adequacy and reliability of the models. These findings validate the accuracy of the mixture design approach used to predict biomass responses. The detailed validation parameters are presented in S6 Table.

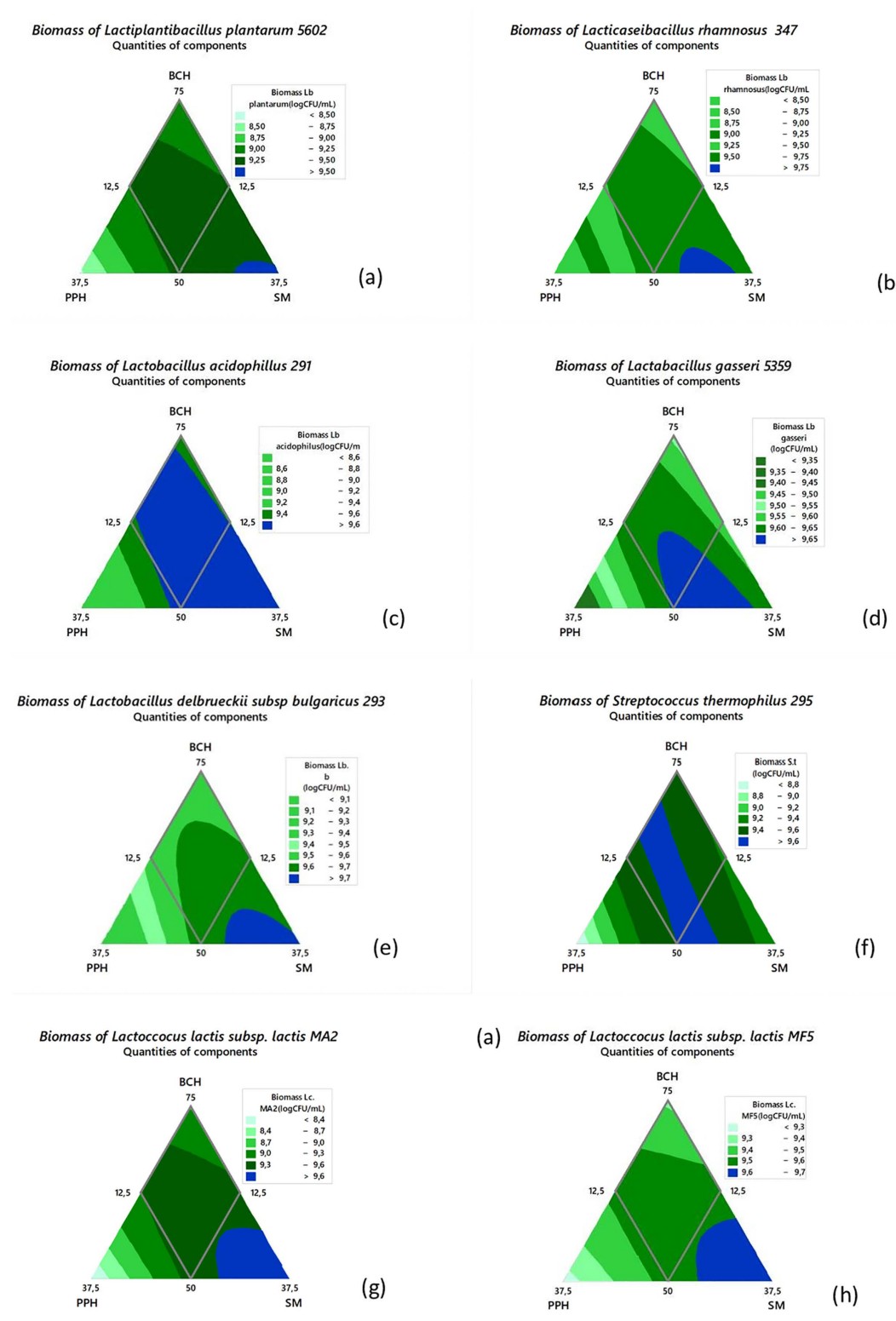

**Fig 1. Iso-response curves showing the optimal estimated proportions of the mixture for the optimal growth of the different strains.** *Lb plantarum* *Lactiplantibacillus plantarum* 5602(a)*, **Lb rhamnosus**: *Lacticaseibacillus rhamnosus* 347(b), **Lb acidophilus:** *Lactobacillus acidophilus* 291(c), **Lb gasseri:** *Lactobacillus gasseri* 5359(d), **Lb b:** *Lactobacillus delbrueckii* subsp. *bulgaricus* 293 (e), **S.t:** *Streptococcus thermophilus* 295 (f); **Lc. MA2:**

*Lactococcus lactis* subsp. *lactis* MA2 (g), **Lc. MF5:** *Lactococcus lactis* subsp. *lactis* MF5 (h), **B. bifidum:** *Bifidobacterium bifidum* 231(i) and **B. subtilis:** *Bacillus subtilis* 215 (j). **BCH:** black soldier fly larvae cake hydrolysate (% v/v); **PPH:** pineapple peel hydrolysate (% v/v); **SM:** 4% sugarcane molasses (%v/v).

## Iso-response curves for biomass production

Fig 1 below shows the different iso-response curves for each measured biomass, illustrating the optimal growth zones for the various strains: *Lactiplantibacillus plantarum* 5602(a), *Lacticaseibacillus rhamnosus* 347(b), *Lactobacillus acidophilus* 291(c), *Lactobacillus gasseri* 5359(d), *Lactobacillus delbrueckii* subsp. *bulgaricus* 293 (e), *Streptococcus thermophilus* 295 (f); *Lactococcus lactis* subsp. *lactis* MA2 (g), *Lactococcus lactis* subsp. *lactis* MF5 (h), *Bifidobacterium bifidum* 231(i) and *Bacillus subtilis* 215(j), according to the different proportions of BC hydrolysate, pineapple peel hydrolysate and a 4% sugarcane molasses solution

The iso-response curves presented in (Fig 1) illustrate the variation in biomass production for the different bacterial strains tested, as a function of the relative proportions of three agro-industrial by-products: black soldier fly larvae cake hydrolysate (BCH), pineapple peel hydrolysate (PPH), and a 4% sugarcane molasses solution (SM).

Each ternary diagram displays contour lines representing equal levels of biomass response, allowing the identification of regions where optimal or suboptimal growth occurs for each strain. For *Lactiplantibacillus plantarum* 5602, the highest biomass values are located in the central region of the triangle, where the three components are balanced. This suggests a synergistic effect between carbon sources (molasses, pineapple) and nitrogen sources (BCH). with reference to *Lacticaseibacillus rhamnosus* 347, optimal biomass is concentrated toward the BCH-SM axis, indicating that *Lacticaseibacillus rhamnosus* 347 benefits from the combined presence of proteins and sugars, while PPH alone appears less effective. When it comes to *Lactobacillus acidophilus* 291, growth is maximal in the region rich in molasses, with moderate contributions from BCH. This suggests a strong preference for readily fermentable sugars, especially glucose and sucrose. As for *Lactobacillus gasseri* 5359, the optimum zone is skewed toward BCH-rich compositions, indicating that this strain is highly dependent on amino acids and peptides provided by the larvae hydrolysate. With regard to *Lactobacillus delbrueckii* subsp. *bulgaricus* 293, this strain reaches peak biomass in molasses-dominated mixtures, consistent with its known metabolic profile which favours high-energy carbon sources. The biomass of *Streptococcus thermophilus* 295 is maximised at intermediate concentrations of SM and BCH, with PPH having a limited effect, whilst the combination of sugars and moderate nitrogen supports this thermophilic strain. The optimal growth zone of *Lactococcus lactis* subsp. *lactis* MA2, lies along the PPH-SM axis, showing a strong affinity for fruit-derived sugars and molasses, what may reflect the strain's adaptation to sugar-rich matrices. Concerning *Lactococcus lactis* subsp. *lactis* MF5, biomass increases toward the centre, suggesting balanced contributions from all three ingredients; the widely spaced iso-response lines indicate high tolerance to formulation shifts. Regarding *Bifidobacterium bifidum* 231, the highest biomass is found near BCH and SM, suggesting a dependency on both protein hydrolysates and fermentable sugars, while PPH has little impact. (j) The peak biomass of *Bacillus subtilis* 215 production occurs in BCH-rich regions, consistent with its proteolytic capacity and lesser dependence on simple sugars. Across all strains, the optimal growth zones are generally located in combinations involving BCH and SM, with PPH playing a supporting role or contributing partially. This reflects the importance of providing both nitrogen (from BCH) and carbon (from SM and PPH) sources in adequate proportions.

## Compromise optimal mixture and validation of the predicted optimal biomass values of the strains

As previously noticed, optimal biomass of the strains was not obtained with one mixture (trial), but four of them (trials 1, 2, 5 and 6). Reason why, it was of utmost importance to define a compromise optimal mixture (optimal proportions of by-products) for obtaining optimal growth of all the strains at once. In this respect, the individual optimal mixtures of the strains were analysed using MINITAB 18 to define the optimal proportions of each by-product required to formulate the

**Table 8. By-product proportions of the compromise medium and validation of biomass production model.**

| Strains | BCH (%v/v) | PPH (%v/v) | SM (%v/v) | Predicted optimal biomass values (log CFU/ mL) | Optimal experimental biomass values (log CFU/ mL) | Desirability |
|---|---|---|---|---|---|---|
| *Lactiplantibacillus plantarum* 5602 | 55.15 | 19.85 | 25 | 9.39±0.10[a] | 9.43±0.55[a] | 0.95 |
| *Lacticaseibacillus rhamnosus* 347 | 55.15 | 19.85 | 25 | 9.70±0.08[a] | 9.77±0.24[a] | |
| *Lactobacillus acidophilus* 291 | 55.15 | 19.85 | 25 | 9.74±0.22[a] | 9.86±0.01[a] | |
| *Lactobacillus gasseri* 5359 | 55.15 | 19.85 | 25 | 9.67±0.15[a] | 9.58±0.09[a] | |
| *Lactobacillus delbrueckii* subsp. *bulgaricus* 293 | 55.15 | 19.85 | 25 | 9.68±0.09[a] | 9.70±0.00[a] | |
| *Streptococcus thermophilus* 295 | 55.15 | 19.85 | 25 | 9.62±0.11[a] | 9.67±0.10[a] | |
| *Lactococcus lactis* subsp. *lactis* MA2 | 55.15 | 19.85 | 25 | 9.51±0.05[a] | 9.47±0.36[a] | |
| *Lactococcus lactis* subsp. *lactis* MF5 | 55.15 | 19.85 | 25 | 9.57±0.10[a] | 9.52±0.40[a] | |
| *Bifidobacterium bifidum* 231 | 55.15 | 19.85 | 25 | 9.61±0.13[a] | 9.65±0.54[a] | |
| *Bacillus subtilis* 215 | 55.15 | 19.85 | 25 | 9.62±0.19[a] | 9.60±0.06[a] | |

[a-a]: In the same line, means with different lower-case letters differ significantly ($p < 0.05$) **BCH:** black soldier fly larvae cake hydrolysate (% v/v); **PPH:** pineapple peel hydrolysate (% v/v); **SM:** 4% sugarcane molasses (%v/v).

**Table 9. Antimicrobial activity of selected LAB strains cultivated in the media formulated based on experimental matrix.**

| Trials | BCH (% v/v) | PPH (% v/v) | SM (% v/v) | *Lactiplantibacillus plantarum* 5602/ *S. aureus* NCDC 109 | *Lacticasebacillus rhamnosus* 347/ *S. aureus* NCDC 109 | *Lactococcus lactis* subsp. *lactis* MA2/ *Pediococcus acidilactici* | *Lactococcus lactis* subsp. *lactis* MF5/ *Pediococcus acidilactici* | *Bifidobacterium bifidum*/ *E. coli* ATCC 11775* |
|---|---|---|---|---|---|---|---|---|
| | | | | Inhibition zone diameter (mm) | | | | |
| 1 | 62.500 | 15.625 | 21.8750 | 10.50±0.28[ed] | 10.00±0.42[dc] | 13.70±0.42[fe] | 13.50±0.14[fe] | 11.00±0.150[de] |
| 2 | 56.250 | 21.875 | 21.875 | 10.50±0.00[ed] | 10.00±0.41[dc] | 13.00±0.17[ed] | 14.00±0.07[f] | 11.00±0.11[de] |
| 3 | 68.750 | 15.625 | 15.625 | 10.00±0.20[dc] | 9.50±0.00[cb] | 12.50±0.7[dc] | 11.50±0.35[b] | 10.50±1.40[c] |
| 4 | 50.000 | 25.000 | 25.000 | 10.50±0.18[ed] | 9.50±0.29[cb] | 13.00±0.4[ed] | 13.00±0.00[ed] | 10.50±0.28[c] |
| 5 | 62.500 | 18.750 | 18.750 | 11.50±0.40[f] | 11.50±0.37[e] | 14.00±0.9[f] | 13.00±0.07[ed] | 11.50±0.17[d] |
| 6 | 62.500 | 12.500 | 25.000 | 10.5±0.70[ed] | 10.70±0.00[ed] | 13.50±0.07[fe] | 13.40±0.4[fe] | 10.50±0.70[c] |
| 7 | 62.500 | 25.000 | 12.500 | 9.00±0.41[b] | 9.00±0.7[b] | 11.50±0.5[b] | 12.00±0.3[cb] | 9.00±0.70[b] |
| 8 | 62.500 | 21.875 | 15.625 | 9.50±0.70[cb] | 9.50±0.7[cb] | 12.00±0.00[cb] | 12.50±0.56[dc] | 9.50±0.00[b] |
| 9 | 75.000 | 12.500 | 12.500 | 7.00±0.00[a] | 7.00±0.07[a] | 9.00±0.07[a] | 10.00±0.07[a] | 8.00±0.42[a] |
| MRS/ M17 | | | | 11.00±0.00[fe] | 11.50±0.27[e] | 14.00±0.4[f] | 14.00±0.00[f] | 11.00±0.37[de] |

*: **Producer**/Indicator. [a-f]: In the same column, means with different lower-case letters differ significantly ($p < 0.05$). Culture on MRS medium: *Lacticaseibacillus rhamnosus* 347, *Lactiplantibacillus plantarum* 5602, *Lactobacillus acidophilus* 291, *Bifidobacterium bifidum* 231, *Lactobacillus delbrueckii* subsp. *bulgaricus* 293, *Lactobacillus gasseri* 5359, *Bacillus subtilis* 215. Culture on M17 medium: *Lactococcus lactis* subsp. *lactis* MA2, *Lactococcus lactis* subsp. *lactis* MF5 and *Streptococcus thermophilus* 295. **BCH:** black soldier fly larvae cake hydrolysate (% v/v); **PPH:** pineapple peel hydrolysate (% v/v); **SM:** 4% sugarcane molasses (%v/v).

compromise optimal medium. Table 8 and S7 Table show that the optimal growth of all the strains requires a compromise medium with 55.1508% (v/v) BCH 19.849% (v/v) PPH and 25% (v/v) of a 4% SM solution.

Based on the compromise mixture, optimal biomass values were predicted by the MINITAB 18 software, and for their confirmation, further growth experiments of each strain in the compromise medium were carried out and the experimental optimal values obtained were compared to the predicted ones, as shown in Table 8. Overall, the experimental optimal biomass values obtained do not differ significantly ($p < 0.05$) from the predicted ones, with a desirability of 0.95, thus validating the biomass values predicted by the software.

## Bacteriocin production in media formulated based on experimental matrix

Table 9 below and S8 Table show the results of the antimicrobial activity tests of the neutralised cell-free supernatants of bacteriocin-producing strains against indicator strains. The inhibition zone diameters recorded for *Bifidobacterium bifidum* 231 against *E. coli* ATCC 11775 for the various trials ranged from 8.00±0.42 to 11.50±0.17 mm, with the significantly (p<0.05) highest value, also significantly higher than that obtained in MRS (11.00±0.37 mm) obtained with trial 5. With regard to *Lactiplantibacillus plantarum* 5602, the recorded inhibition zone diameters against *Staphylococcus aureus* NCDC 109 for the various trials ranged from 7.00±0.00 to 11.50±0.40 mm, with the significantly (p<0.05) highest value, also significantly (p<0.05) higher than that obtained in MRS (11.00±0.00 mm) obtained with trial 5. Concerning *Lacticaseibacillus rhamnosus* 347, the recorded inhibition zone diameters against *Staphylococcus aureus* NCDC 109 for the various trials ranged from 7.00±0.07 to 11.50±0.37 mm, with the significantly(p<0.05) highest value equal to that obtained in MRS (11.50±0.27 mm) obtained with trial 5. As for *Lactococcus lactis* subsp. *lactis* MA2, the inhibition zone diameters recorded against *Pediococcus acidilactici* for the various trials ranged from 9.00±0.07 to 14.00±0.90 mm, with the significantly (p<0.05) highest value equal to that obtained in MRS (14.00±0.40 mm) obtained with trial 5. With reference to *Lactococcus lactis* subsp. *lactis* MF5, the inhibition zone diameters recorded against *Pediococcus acidilactici* for the various trials ranged from 10±0.07 to 14.00±0.07 mm, with the significantly (p<0.05) highest value equal to that obtained in MRS (14.00±0.00 mm) obtained with trial 2.

Table 10 presents the antimicrobial activities of bacteriocin-producing strains cultivated in the formulated low-cost media compared to the control (MRS/M17). Overall, the formulated media supported comparable or increased antimicrobial activity relatively to the control. Strains showing increased activity (p<0.05) than in the control are *Lactiplantibacillus plantarum* 5602 *and Bifidobacterium bifidum* 231. Strains with comparable activity to that in the control are *Lactococcus lactis* subsp. *lactis* MF5, *Lactococcus lactis* subsp. *lactis* MA2 and *Lacticaseibacillus rhamnosus* 347.

## Proposed mathematical models for Bacteriocin production and contribution of factors

After analysing the effect of each proportion as well as the interactions of the by-products (proportions of BC hydrolysate, pineapple peel hydrolysate and sugarcane molasses solution) on the response (bacteriocin production), the mathematical model for each strain was predicted using the following regression equations:

Antimicrobial activity of *Lactiplantibacillus plantarum* 5602

$$-0.118x1 - 2.37x2 + 0.27x3 + 0.043x1x2 + 0.02x2x3 = Y11 \tag{14}$$

Antimicrobial activity of *Lacticaseibacillus rhamnosus* 347

$$-0.131x1 - 2.153x2 + 0.38x3 + 0.038x1x2 + 0.019x2x3 = Y12 \tag{15}$$

**Table 10. Comparative antimicrobial activities of the strains in the formulated and control (MRS/M17) media.**

| Strains | Activity range (mm) | Max activity value in formulated media (mm) | Activity value in Control media (mm) | p-value | Comparative effect |
|---|---|---|---|---|---|
| *Lactiplantibacillus plantarum* 5602 | 7.00±00 −11.50±04 | 11.50±0.40 | 11.00±0.0 | 0.031 | Superior |
| *Lacticaseibacillus rhamnosus* 347 | 7.00±0.07-11.50±0.37 | 11.50±0.37 | 11.50±0.27 | 0.639 | Comparable |
| *Lactococcus lactis* subsp. *lactis* MA2 | 9.00±0.07-14.09±0.40 | 14.09±0.40 | 14.00±0.40 | 0.779 | Comparable |
| *Lactococcus lactis* subsp. *lactis* MF5 | 10.0 0±0.07-14.00±0.07 | 14.00±0.07 | 14.00±0.00 | 0.834 | Comparable |
| *Bifidobacterium bifidum* 231 | 8.00±0.42-11.5±0.17 | 11.50±0.17 | 11.00±0.37 | 0.042 | Superior |

p<0.05 was considered significant.

Antimicrobial activity of *Lactococcus lactis* subsp. *lactis* MA2

$$-0.106x1 - 2.3x2 + 0.398x3 + 0.041x1x2 + 0.02x2x3 = Y13 \qquad (16)$$

Antimicrobial activity of *Lactococcus lactis* subsp. *lactis* MF5

$$-0.062x1 - 1.274x2 + 0.419x3 + 0.025x1x2 + 0.009x2x3 = Y14 \qquad (17)$$

Antimicrobial activity of *Bifidobacterium bifidum* 231

$$-0.08x1 - 2.51x2 + 0.16x3 + 0.04x1x2 + 0.036x2x3 = Y15 \qquad (18)$$

where: **Y**=Biomass (log CFU/mL); **x1**=Black soldier fly larvae cake hydrolysate (% v/v); **x2**=Pineapple peel hydrolysate (% v/v); **x3**=Sugarcane molasses (% v/v).

The analysis of the regression equations highlights the respective contributions of linear effects and interactions among the media constituents on the antimicrobial activity of the tested strains. The linear effect of SM($x_3$), along with its interaction of BC hydrolysate and pineapple peel hydrolysate ($x_1x_2$) and pineapple peel and molasses hydrolysates ($x_2x_3$) contributed positively to the antimicrobial activity of all tested strains, namely *Lactiplantibacillus plantarum* 5602, *Lacticaseibacillus rhamnosus* 347, *Lactococcus lactis* subsp. *lactis* MA2 and MF5 and *Bifidobacterium bifidum* 231, highlighting the significance of these factors in optimising antimicrobial activity. In contrary, the linear effects of BCH ($x_1$) and PPH ($x_2$) were negative on bacteriocin production.

### Validation of mathematical models for bacteriocin production

The coefficient of determination ($R^2$) for the proposed models ranged from 85.82% to 96.26%, the absolute mean deviation analysis (AMDA) ranged from 0.00 to 0.02 and the bias factor (Bf) range was 0.99 to 1.05 (S9 Table). All the values of these parameters fall within the acceptable reference range to validate a model ($R^2$ are ≥ 0.75; AMDA=0; 0.75<Bf<1.25).

### Iso-response curves for different responses for bacteriocin production

The iso-response curves corresponding to each measured response illustrate the optimal inhibition zones for the different strains, namely *Lactiplantibacillus plantarum* 5602(a*), Lacticaseibacillus rhamnosus* 347(b); *Lactococcus lactis* subsp. *lactis* MA2 (c*), Lactococcus lactis* subsp. *lactis* MF5 (d) *Bifidobacterium bifidum* 231(e). these results are presented as a function of varying proportions from black soldier fly larvae cakes hydrolysate (BCH), pineapple peel hydrolysate (PPH) and 4% sugarcane molasses (SM) solution, as shown in (Fig 2):

It reveals that each ternary diagram includes contour lines representing equal inhibition levels, making it possible to identify the conditions that promote either strong or weak antagonistic effects for each bacterial strain. For *Lactiplantibacillus plantarum* 5602 (a), the most pronounced inhibitory effect is observed in the central region of the triangle, where the three components are balanced. This suggests a synergistic interaction between carbon-rich (PPH, SM) and nitrogen-rich (BCH) substrates in enhancing bacteriocin production or other inhibitory metabolites. In contrast, *Lacticaseibacillus rhamnosus* 347 (b) showed maximal antimicrobial performance along the BCH-SM axis, indicating that this strain benefits particularly from the combination of protein and sugar sources. The use of PPH alone appears to be less effective in stimulating inhibitory activity. As for *Lactococcus lactis* subsp. *lactis* MA2 (c), optimal inhibition zones are located along the PPH-SM axis, reflecting a strong response to fruit-derived sugars and molasses, and possibly an adaptation of this strain to sugar-enriched environments for bacteriocin synthesis. Regarding *Lactococcus lactis* subsp. *lactis* MF5 (d), the wide space between contour lines indicates a broad tolerance to changes in formulation, suggesting stable antimicrobial

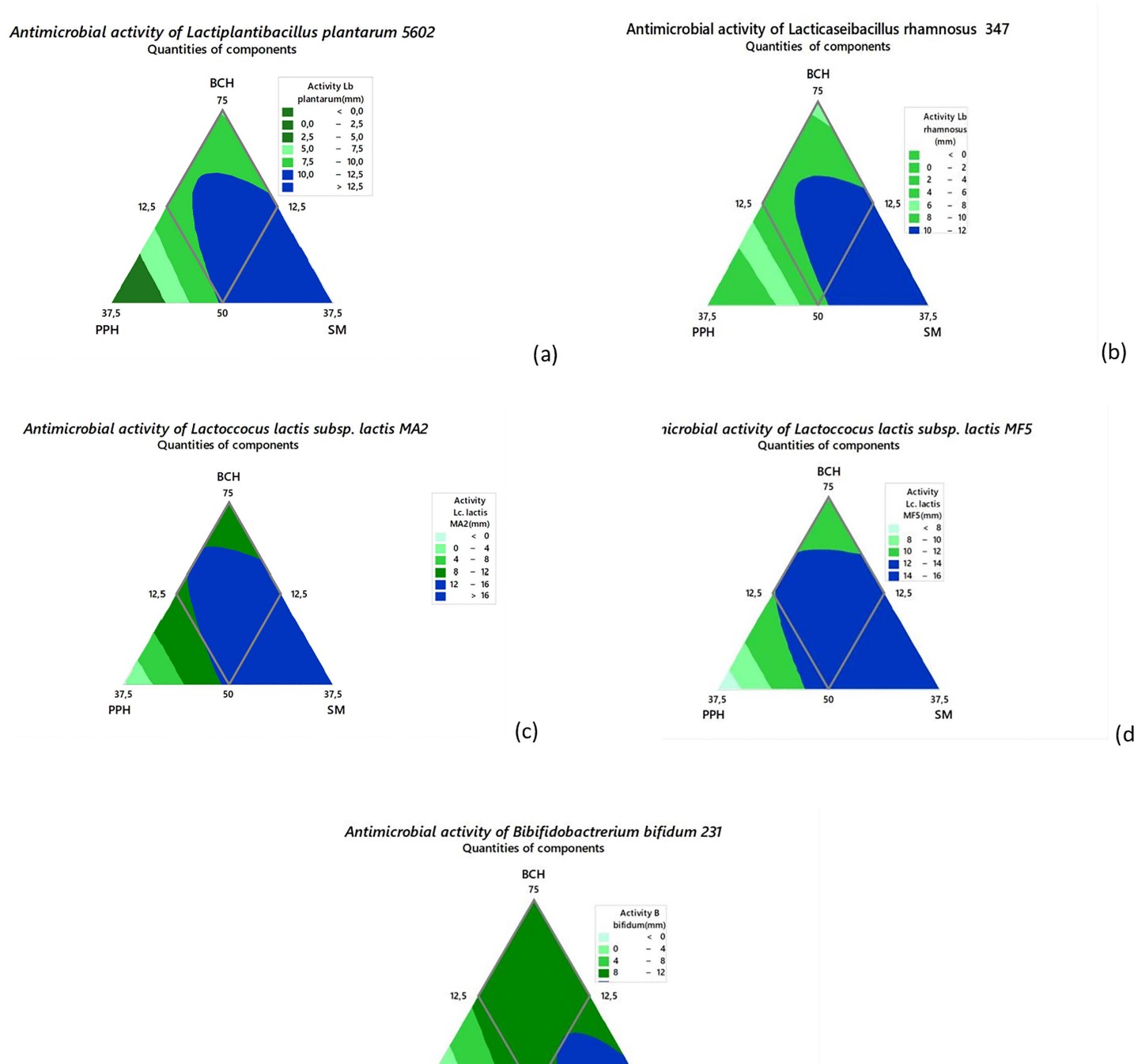

**Fig 2. Iso-response curves showing the optimal estimated proportions of the mixture for the optimal bacteriocin production by different strains.** *Lb plantarum* Lactiplantibacillus plantarum 5602 (a); ***Lb rhamnosus:*** *Lacticaseibacillus rhamnosus* 347 (b); ***Lc.* MA2:** *Lactococcus lactis* subsp. *lactis* MA2 (c); ***Lc.* MF5:** *Lactococcus lactis* subsp. *lactis* MF5 (d) and ***B. bifidum:*** *Bifidobacterium bifidum* (e) 231. **BCH:** black soldier fly larvae cake hydrolysate (% v/v); **PPH:** pineapple peel hydrolysate (% v/v); **SM:** 4% sugarcane molasses (%v/v).

performance across different compositions. Finally, *Bifidobacterium bifidum* 231 (e) exhibits optimal inhibition between PPH-SM, highlighting the key role of PPH and SM derived sugars in enhancing its antimicrobial potential.

**Validation of predicted and experimental values of optimal bacteriocin production based on the compromise mixture**

The optimum values predicted by the MINITAB 18 software and experimental values for each trial of the medium formulated under optimum conditions are presented in Table 11 below as well as S10 Table. Briefly, the values obtained after experimentation do not differ significantly (P<0.05) from the predicted results with a desirability of 0.95, thus validating the optimal conditions defined by the software.

**Equation to establish the formula of the compromise culture medium**

The formula of the compromise medium was defined based on the macronutrients composition of the by-products (BCH, PPH and 4% SM). The culture medium formula is based on the macronutrient composition of the various hydrolysates and by-product preparations. Table 12 below and S11 Table show the macronutrient composition of the various by-product hydrolysates.

Based on the formula in equations (1), (2) and (3), the formula of the medium was defined as:

$$10(3.16X + 1.98Y + 0.26Z) = P \tag{19}$$

$$10(0.08X + 0.15Y + 0.00Z) = L \tag{20}$$

$$10(3.47X + 5.31Y + 3.47Z) = C \tag{21}$$

**Table 11. By-products proportions of the compromise medium and validation of bacteriocin production model.**

| Bacteriocin production | BCH (%v/v) | PPH (%v/v) | SM (%v/v) | Predicted optimal activity values (mm) | Experimental optimal activity values (mm) | Desir-ability |
|---|---|---|---|---|---|---|
| *Lactiplantibacillus plantarum* **5602** | 55.15 | 19.85 | 25 | 11.83±0.10[a] | 11.70±0.21[a] | 0.95 |
| *Lacticaseibacillus rhamnosus* **347** | 55.15 | 19.85 | 25 | 11.40±0.05[a] | 11.70±0.1[a] | |
| *Lactococcus lactis* subsp. *lactis* **MA2** | 55.15 | 19.85 | 25 | 14.70±0.11[a] | 14.50±0.21[a] | |
| *Lactococcus lactis* subsp. *lactis* **MF5** | 55.15 | 19.85 | 25 | 14.32±0.08[a] | 14.10±0.11[a] | |
| *Bifidobacterium bifidum* **231** | 55.15 | 19.85 | 25 | 12.00±0.02[a] | 11.90±0.13[a] | |

**BCH:** black soldier fly larvae cake hydrolysate (% v/v); **PPH:** pineapple peel hydrolysate (% v/v); **SM:** 4% sugarcane molasses (%v/v).

a-a In the same line, means with different lower-case letters differ significantly (p<0.05).

**Table 12. Macronutrients composition of BC hydrolysate, pineapple hydrolysate and 4% sugarcane molasses preparation.**

| By-products | Proteins (%DM) | Lipids (%DM) | Carbohydrates (% DM) | Ash (%DM) |
|---|---|---|---|---|
| BCH | 3.16±0.03 | 0.08±0.05 | 3.47±0.07 | 1.13±0.10 |
| PPH | 1.98±0.04 | 0.15±0.08 | 5.31±0.03 | 0.34±0.03 |
| 4% SM | 0.28±0.17 | 0.00±0.00 | 3.47±0.05 | 0.24±0.05 |

**BCH:** black soldier fly larvae cake hydrolysate; **PPH:** pineapple peel hydrolysate; **SM:** 4% sugarcane molasses.

Where: **X** = dry matter content (g) in 55.1508 mL of black soldier fly larvae cake hydrolysate; **Y** = dry matter content (g) in 19.849 mL of pineapple peel hydrolysate; **Z** = dry matter content (g) in 25.00 mL of 4% sugarcane molasses

Using this formula, the final macronutrient contents (g/L) in the medium was calculated and found to be: 14.99 g/L of proteins, 0.80 g/L of lipids, and 24.67 g/L of carbohydrates.

## Discussion

### Determination of the physico-chemical composition of the by-products

The physico-chemical analysis of pineapple peels revealed contents of 0.75% lipids, 4.89% protein, and 89.52% carbohydrates. These values are significantly higher than those reported by Santos et al. [16], who observed 0.5%, 3%, and 60% respectively, in Brazilian samples. Similarly, Thatchajaree et al. [28] reported 2.53% lipids, 4.20% protein, and 73.30% carbohydrates in dried pineapple peels from Thailand. Such variations are likely due to factors including geographical origin, agro-ecological conditions, and post-harvest processing techniques [29]. Beyond their carbohydrate richness, pineapple peels also displayed a noteworthy amino acid profile. The analysis revealed a predominance of lysine, followed by proline, aspartic acid, glycine, alanine, histidine, serine, glutamic acid, and cysteine. These amino acids, particularly lysine, are essential for the growth of several lactic acid bacteria (LAB), such as *Lactobacillus acidophilus* and *Bifidobacterium bifidum*, which are auxotrophic for this compound. The presence of both sugars and amino acids in this by-product makes it a dual-purpose ingredient in culture medium formulation. As for sugarcane molasses, the results indicated 0% lipids, 7.02% protein, and a high total sugar content of 86.86%. These values differ from those reported by Feumba et al. [29], who found 1% lipids, 6% protein, and 65% sugars. Such discrepancies may stem from differences in sugar extraction processes, as well as pedo-climatic and cultivation factors [30]. The sugar profile was dominated by sucrose (579.38 ± 2.40 g/100 g), followed by glucose (195.79 ± 0.10 g/100 g) and fructose (112.47 ± 0.5 g/100 g). These fermentable sugars are widely recognised as readily assimilable carbon sources for LAB such as *Lb. plantarum*, *Lb. acidophilus,* and *B. bifidum*. Thus, molasses serves as a key source of fermentable carbon in the tested formulations. Simultaneously, the BC showed 15% lipids, 48% protein, and 26.27% total sugars. These findings vary slightly from those of Ahmad et al. [15] who reported lipid contents ranging from 10 to 27.65%, protein levels at 55%, and sugars between 10 and 20%. Similarly, Christian et al. [31] reported 25.69 ± 0.12% lipids and 48.20% proteins. Such differences are likely attributable to the larvae's diet, rearing conditions (temperature, humidity), and physiological stage at harvest [32]. The amino acid profile of this ingredient revealed the presence of 17 amino acids, dominated by histidine, glutamic acid, aspartic acid, glycine, and alanine. These results are consistent with those of Christian et al. [31] who identified 18 amino acids with a similar composition. These compounds play an essential role in protein synthesis and metabolic regulation in various probiotic strains, notably *Lb. gasseri*, *S. thermophilus,* and *Lb. bulgaricus,* which exhibit multiple amino acid requirements. To optimise the formulation, a simplex-centroid mixture design was applied. This experimental approach allowed the modeling of combined ingredient effects on the biological response (biomass production). The resulting iso-response curves highlighted positive interactions among the tested components. Notably, the central region of the mixture triangle corresponding to balanced proportions was frequently associated with optimal responses. This suggests a synergistic effect, in which the blend of simple sugars (from molasses and pineapple peels) and amino acids (mainly from larvae cake) provides a nutritionally favorable environment for LAB growth. This observation aligns with findings by Oonincx et al. [33] who used a mixture design to optimise the growth of *Latilactobacillus sakei* using various food-derived nitrogen sources. Their study demonstrated that specific combinations yielded better outcomes than any individual component alone. Similar conclusions were drawn by Galante et al. [34] and Cárdenas et al. [35] who confirmed the relevance of this method for optimising agro-industrial by-product-based media, particularly by capturing complex nutritional interactions. The combined use of pineapple peels, sugarcane molasses, and BC provides a complementary supply of fermentable sugars, proteins, and essential amino acids, meeting the nutritional requirements of lactic acid and probiotic bacteria. The mixture design approach enabled the identification of optimal proportions and demonstrated significant nutritional synergy, validating the potential of these agro-industrial by-products as effective, sustainable alternatives to conventional culture media

## Development of the by-products-based low-cost culture medium

**Performances of the formulated medium on biomass production.** In this study, nine strains of lactic acid bacteria (LAB) and one non-LAB strain with probiotic and technological interest were evaluated for their growth on a culture medium formulated from agro-industrial by-products. After 48 hours of incubation, high cell viabilities were observed, demonstrating that the formulated medium, enriched by enzymatic hydrolysis of by-products, provides an optimised environment for bacterial growth. In some cases, growth was even higher than that obtained with conventional commercial media such as MRS and M17. The high biomass levels reached (between 9.04 and 9.73 log CFU/mL) highlight the effectiveness of combining hydrolysed by-products. The enzymatic hydrolysis of pineapple peels and BC released fermentable sugars, peptides, proteins, amino acids, polyphenols, and minerals, thus increasing nutrient bioavailability for the bacteria. Previous studies have reported that enzymatic or acid hydrolysis improves the digestibility and accessibility of substrates, thereby promoting microbial growth [36]. Notably, strains such as *Lacticaseibacillus rhamnosus* and *Lactiplantibacillus plantarum* have demonstrated efficient metabolism of pineapple peel hydrolysates [37]. Furthermore, diluted molasses has been identified as an excellent substrate for growth enhancement in LAB [38]. The positive linear effects of BC hydrolysate and molasses could be explained by their nutritional contents, as they provide carbon, nitrogen and minerals requirements for bacterial growth. The positive interactions between BC hydrolysate and molasses on the one hand, and pineapple peel hydrolysate and molasses on the other hand suggest a complementary of synergism in the nutrients provided by each source. These patterns agree with previous observations that the synergistic balance between nitrogen-rich and carbohydrate-rich by-products improves LAB performance in mixed-substrate media [39]. The mixture design method was successfully applied in recent microbial culture medium formulations. This approach not only enables a cost-effective alternative to commercial media but also fits within the principles of a circular economy by valorising agro-industrial by-products. It simplifies laboratory and industrial protocols while respecting the strict nutritional requirements of lactic acid bacteria [40].

**Performances of the formulated medium on bacteriocin production.** Bacteriocins are characterised by strong auxotrophic requirements due to their limited ability to synthesise certain vitamins and amino acids. Their growth is only possible in rich and complex culture media containing mainly: a nitrogen source (peptides, amino acids), a carbon source, vitamins, and minerals. These nutrients must be provided in optimal concentrations [41]. Five bacteriocin-producing lactic acid bacteria strains were studied, the antimicrobial activity test of the bacteriocin-producing strains showed inhibition diameters of 11 mm for *Lactiplantibacillus plantarum* 5602, 11.5 mm for *Lacticaseibacillus rhamnosus* 347, 14 mm for *Lactococcus lactis* subsp. *lactis* MA2, 14 mm for *Lactococcus lactis* subsp. *lactis* MF5, and 11 mm for *Bifidobacterium bifidum* 231. All strains exhibited equal or higher activity in the formulated medium compared to MRS/M17 media. This could be attributed to their ability to efficiently metabolise the sugars and amino acids present in the hydrolysates of pineapple peels and BC. These results are consistent with those of Dabrowska et al. [11] who demonstrated that the addition of hydrolysed proteins to a culture medium improves bacterial growth and bacteriocin production by providing a more accessible nitrogen source and, in some cases, increased bioactivity. Similarly, Abbasiliasi et al. [42] showed that some probiotic strains enhance biomass production yield when hydrolysates are added to the formulated medium. The good performance of the formulated medium can also be explained by its richness in essential amino acids such as lysine, leucine, and valine mainly derived from BC, which is recognised for its high protein content and well-balanced amino acid profile [43,44]. Pineapple peels and sugarcane molasses additionally provide fermentable sugars and minerals, creating a complete and highly bioavailable nutritional environment. These components are essential for cell growth and the biosynthesis of antimicrobial compounds such as bacteriocins. The results of this study thus confirm the relevance of these agro-industrial by-products in formulating low-cost culture media for bacterial biomass and bacteriocin production. Bacteriocin production depends on the cell density and metabolic activity of lactic acid bacteria. It has been shown that bacteriocin synthesis is often correlated with bacterial growth, particularly during the exponential phase when cells are metabolically active [41]. The availability of nitrogen and other nutrients such as proteins and sugars influenced both growth and production of antimicrobial metabolites [45].

## Compromise medium and its formula

The compromise medium formulated contains 55.15 mL of BC hydrolysate, 19.84 mL of pineapple peel hydrolysate, and 25 mL of 4% sugarcane molasses solution per 100 mL of final medium. This medium offers a balanced composition of essential macronutrients, notably proteins, lipids, and carbohydrates. It is particularly rich in simple sugars such as glucose and fructose, which are rapidly metabolised by lactic acid bacteria, promoting an initial fast growth phase. Moreover, the presence of more complex sugars like sucrose allows a gradual release of carbohydrate sources, ensuring a sustained energy supply during fermentation. This sequence of sugar utilisation is supported by Corsetti et al. [46] who highlighted that lactic acid bacteria preferentially consume simple sugars before progressively metabolising complex sugars, thus optimising their growth and metabolic activity. The lipids present in this medium, although at low concentrations, play a crucial role in membrane structure and protection against environmental stress. According to Srinivas et al. [47] lipids contribute to cell membrane stability and enhance the resistance of lactic acid bacteria to stress conditions such as freeze-drying and other adverse environments, thereby improving the viability and productivity of the cultures. Proteins derived mainly from the BC provide a significant source of amino acids. This substrate contains 17 essential and non-essential amino acids, offering a comprehensive nutrient profile for lactic acid bacteria. The availability of these amino acids is critical since growth factor requirements vary considerably among bacterial strains. Indeed, Zhao et al. [48] demonstrated that lactic acid bacteria differ in their capacity to produce and utilise extracellular amino acids, implying that each strain has specific nutritional needs. This variability underscores the importance of a custom-formulated culture medium capable of meeting the specific preferences of microorganisms to optimise their growth and bacteriocin production. Thus, the medium formulated from agro-industrial by-products not only provides a complete nutritional composition but also presents a resource utilisation dynamic adapted to the metabolic requirements of lactic acid bacteria. This formula results in good bacterial growth and bacteriocin production performances while valorising local, economical, and sustainable resources.

## Conclusion

This study was aimed to formulate a low-cost culture medium able to sustain the growth and beneficial metabolites production of a wide genera of lactic acid bacteria, from agro-industrial and environmental by-products.

Based on their nutrient contents, black soldier fly larvae cake, pineapple peels and sugarcane molasses were found to be good nitrogen and carbon sources for LAB growth and bacteriocin production. The use of experimental mixing design made it possible to obtain optimal proportions of the ingredients in the medium as 55.15% larvae cake hydrolysate, 19.85% pineapple peel hydrolysate, and 25.00% of 4% sugarcane molasses solution. A compromise medium that optimally sustain the growth and bacteriocin production of bacterial strains from various genera and species, as well as with various purposes (technological properties, probiotics and bacteriocin production) was derived. Its formula was also defined, so as to ease its preparation using these ingredients, but from different origins. Interestingly, the formulated compromise medium performed more to similar than MRS or M17 for strains cultivation and bacteriocin production. By valorising industrial by-products, this study fits into a circular economy and sustainable development model. It opens promising perspectives for the use of this medium at lab-scale for research, or at pilot and industrial-scales (agri-food, pharmaceutical, and cosmetic) for the production of the biomass of lactic starters or probiotics, and bacteriocins.

## Supporting information

**S1 Table. Physico-chemical composition of BC, pineapple peel and sugarcane molasses (Dataset).**
(PDF)

**S2 Table. Amino acids profiles of the various by-products (BC, pineapple peel and sugarcane molasses).**
(PDF)

**S3 Table. Amino acids profiles of the by-products (BC, pineapple peel, sugarcane molasses) (Dataset).**
(PDF)

**S4 Table. Sugar profiles of the various by-products (BC, pineapple peels, sugarcane molasses) (Dataset).**
(PDF)

**S5 Table. Strains' biomass production in the media formulated based on experimental matrix (Dataset).**
(PDF)

**S6 Table. Validation of mathematical models for biomass production.**
(PDF)

**S7 Table. By-products proportions of the compromise medium and validation of biomass production model (Dataset).**
(PDF)

**S8 Table. Strains' bacteriocin production in the media formulated based on experimental matrix (Dataset).**
(PDF)

**S9 Table. Validation of mathematical models for bacteriocin production.**
(PDF)

**S10 Table. By-products proportions of the compromise medium and validation of bacteriocin production model (Dataset).**
(PDF)

**S11 Table. Macronutrients composition of BC hydrolysate, pineapple peel hydrolysate and sugarcane molasses (Dataset).**
(PDF)

## Acknowledgments

The authors thank the CV Raman Fellowship for African Researches, for supporting a research stay in India during which analyses of amino acid profile, sugar profile, and mineral content of the by-products were performed.

## Author contributions

**Conceptualization:** Michele Letitia Tchabou Tientcheu, Pierre Marie Kaktcham , François Zambou Ngoufack.

**Data curation:** Pierre Marie Kaktcham , Edith Marius Foko Kouam, Laverdure Tchamani Piame, Singh Bhim Pratap.

**Formal analysis:** Michele Letitia Tchabou Tientcheu, Edith Marius Foko Kouam, Lysette Chabrone Djodjeu Kamega, Agnihotri Shekhar, Singh Bhim Pratap.

**Investigation:** Michele Letitia Tchabou Tientcheu, Pierre Marie Kaktcham.

**Methodology:** Michele Letitia Tchabou Tientcheu, Edith Marius Foko Kouam, Laverdure Tchamani Piame, Aarzoo, Agnihotri Shekhar, Singh Bhim Pratap.

**Resources:** Lysette Chabrone Djodjeu Kamega.

**Software:** Edith Marius Foko Kouam.

**Supervision:** François Zambou Ngoufa Aarzoo Aarzoo ck.

**Validation:** Pierre Marie Kaktcham, Singh Bhim Pratap.

**Visualization:** Pierre Marie Kaktcham, Laverdure Tchamani Piame, Aarzoo.

**Writing – original draft:** Michele Letitia Tchabou Tientcheu, Lysette Chabrone Djodjeu Kamega.

**Writing – review & editing:** Pierre Marie Kaktcham, Laverdure Tchamani Piame, Aarzoo, Agnihotri Shekhar, Singh Bhim Pratap, François Zambou Ngoufack.

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
