## [Decision Letter · Decision Letter 0]

29 Oct 2025

Dear Dr. Pierre Marie,

Thank you for submitting your manuscript to PLOS ONE. After careful consideration, we feel that it has merit but does not fully meet PLOS ONE’s publication criteria as it currently stands. Therefore, we invite you to submit a revised version of the manuscript that addresses the points raised during the review process.

We look forward to receiving your revised manuscript.

Kind regards,

Guadalupe Virginia Nevárez-Moorillón, Ph.D.

Academic Editor

PLOS ONE

Journal Requirements:

3. We note you have included a table to which you do not refer in the text of your manuscript. Please ensure that you refer to Table 9 in your text; if accepted, production will need this reference to link the reader to the Table.

Reviewers' comments:

Reviewer's Responses to Questions

**Comments to the Author**

1. Is the manuscript technically sound, and do the data support the conclusions?

Reviewer #1: Yes

Reviewer #2: Yes

2. Has the statistical analysis been performed appropriately and rigorously?

Reviewer #1: Yes

Reviewer #2: Yes

3. Have the authors made all data underlying the findings in their manuscript fully available?

Reviewer #1: Yes

Reviewer #2: Yes

4. Is the manuscript presented in an intelligible fashion and written in standard English?

Reviewer #1: Yes

Reviewer #2: Yes

Reviewer #1: 1. The introduction is well-written but repeats general LAB information that is already well-known. This can be shortened, and more space can be dedicated to gaps in existing low-cost media studies.

2. Grammatical Refinement: Minor grammatical errors occur, such as article misuse (“a medium” vs. “the medium”), which need correction.

3. Avoid Repetition: The phrase “black soldier fly larvae cake” is repeated very frequently—consider abbreviation (e.g., BSFLC) after first mention.

4. Why does the author only consider growth measurements by plate counts (While CFU/mL) is valid, it does not provide insights into growth kinetics or substrate utilization clearly. But inclusion of growth curves (OD600 vs. time), pH reduction over time, or metabolite production (e.g., lactic acid concentration) for better assessment of medium performance.

5. In the Result Section Table 5 Amino Acids Profile of the Various By-products - While the amino acid profile of black soldier fly larvae cake, pineapple peels, and molasses is scientifically relevant, the paper’s main focus is on medium performance (growth and bacteriocin activity) rather than nutrient profiling. The amino acid data are supportive but not critical for understanding the experimental outcomes.

6. Table 8 – Validation of Mathematical Models for Biomass Production - This table contains R², AMDA, and bias factor (Bf) values for the regression models. While important for statistical validation, these are technical details that could be summarized in the main text (e.g., “All models had R² > 0.83 and Bf within 0.75–1.25, indicating well fit”). The full table could go to the supplementary section to declutter the results.

Reviewer #2: The following comments are to improve the quality of your research paper:

Introduction section:

Describe more the characteristics or importance of the 3 different sources of the culture

medium. In abbreviations, the name or meaning is placed first, and then the corresponding

abbreviation in parentheses, and you are interchanged in line 44

Materials and methods section:

Check the names of the strains because they are misspelled, for example, Lactoccocus is

Lactococcus, also the term Mc Farland is McFarland, an example is also in line 97, and in

the tables, check each one well.

Improve in general, all the titles of the tables so that they are more descriptive and

punctual, because some are too long, and also check the titles of the columns, because

some have a capital initial and others do not

In Table 1 the term status is the function as described in the title, and the case of

technological is the application of the strain

Try to simplify the methodologies for example autoclaved (121°C, 15psi), in the sequence

of the pineapple peel hydrolysate, the wording is confusing, there is a lack of volume of

water that is used, there is a lack of filters, check also in the black soldier fly larvae cake

the equipment is missing team names, volumes and in the solutions the name of the

reagent is placed first and then the concentration

Check the number of subtitles. The black soldier fly larvae cake is reserved for later use. It

is not specified how they keep it, and they do not mention anything from the number 3

source. Try to be more orderly in writing, follow a sequence in terms of the components of

the medium throughout the article try to handle that sequence to describe them

In line 112, reference is made to the IUPAC and AOAC methods for determining the

nutritional profile of the sample, but it is poorly worded to mention what parameters were

measured and with what technique, respectively, because it remains as an idea in the air.

Because next comes the quantification of minerals, but it is by another type of analysis,

according to the wording

Line 125 goes first with the name of the acid, followed by the concentration

Table 2: Homogenize the initial in all column titles

Improve the wording of the title of table 3, it is very long, you can use a table note or better

describe it in the textLine 180 error in reagent name

In design, it is a mixture design or factorial design

Results section

It is recommended to divide the information into more concise sentences, grouping results

by category, for example, maximum values compared to the control. In the text, use

different words for significantly, use variants such as notably superior, markedly increased,

etc., or place it in the tables of results or methods section.

It is suggested to create a table with the strain, range (CFU/ml), max vs control, p-value,

and write a paragraph that highlights only the key findings to make comparisons clearer in

the text

Check again in this section the names of the strains and units

Explain a little more what the coefficients mean; they only mention linear effects and

interactions. This can help in the interpretation to see if there is inhibition or stimulation

The note in Table 9 is incomplete

Discussion Section:

The section presents a logical structure and supported argumentation, however there are

disconnections with the results and superficial interpretations, for example the result is

described but in the discussion the acceptable rank is not specified if it is or not, if it is

better or not without comparative values, in the terms more similar vs equal or higher there

is contradiction when reading them, In the interpretation of mathematical models as

mentioned before there is no interpretation of the coefficients to carry out the discussion in

this section. The section is totally positive, but what happens with the non-optimal results,

because it can be this result, the variability between replicates, the stability of the

environment at room temperature, or useful life, with its results, it could be scaled, the cost-benefit

**Do you want your identity to be public for this peer review?** For information about this choice, including consent withdrawal, please see our Privacy Policy

Reviewer #1: No

Reviewer #2: No

---

## [Author Response · Author response to Decision Letter 1]

7 Nov 2025

Rebuttal Letter

Article "Development of a low-cost culture medium from industrial and environmental by-products for sustainable cultivation of Lactic Acid Bacteria"

Dear Academic Editor and Reviewers,

We would like to sincerely thank the Editor and both Reviewers for their valuable time, constructive comments, and insightful suggestions. These remarks have greatly helped us to improve the clarity, depth, and scientific quality of our manuscript. Below, we provide a detailed, point-by-point response to each comment, preceded by an R:.

Journal Requirements

1. Please ensure that your manuscript meets PLOS ONE's style requirements, including those for file naming. The PLOS ONE style templates can be found at https://journals.plos.org/plosone/s/file?id=wjVg/PLOSOne_formatting_sample_main_body.pdfandhttps://journals.plos.org/plosone/s/file?id=ba62/PLOSOne_formatting_sample_title_authors_affiliations.pdf

R: The manuscript has been adapted to meet the PLOS ONE’s formatting standards, following the provided templates for structure, tables, and figure captions.

R: Thanks for your observation. To comply with the Data Availability Statement, all the raw data required to reproduce the results (including means, and standard deviations) have been uploaded as Supporting Information files (S4–S11 Tables).

3. We note you have included a table to which you do not refer in the text of your manuscript. Please ensure that you refer to Table 9 in your text; if accepted, production will need this reference to link the reader to the Table.

R: We thank the reviewer for this observation. This concern was addressed in the revised manuscript while the initial Table 9 has become Table 8 and referred in the text.

See Lines 433 – 436_Revised manuscript unmarked: “compared to the predicted ones, as shown in Table 8. Overall, the experimental optimal biomass values obtained do not differ significantly (p˂0.05) from the predicted ones, with a desirability of 0.95, thus validating the biomass values predicted by the software.

Table 8. By-product proportions of the compromise medium and validation of biomass production model”

R: None of the reviewers recommended to cite specific previously published works.

Reviewer #1

1. The introduction is well-written but repeats general LAB information that is already well-known. This can be shortened, and more space can be dedicated to gaps in existing low-cost media studies.

R: The introduction has been revised to reduce general background on lactic acid bacteria and to highlight research gaps and recent advances in low-cost media development. New references have been added (Acosta-Piantini et al. (2023) and Valle Vargas et al., 2024) to emphasize on the sustainability-oriented strategies in medium formulation.

See Line 65 – 69_Revised manuscript unmarked: “Similarly, Valle Vargas et al. [13] demonstrated that media formulated from whey, sugarcane molasses, and palm kernel cake could support the growth of fish probiotics Lactococcus lactis A12, Priestia megaterium M4, and Priestia sp. M10, while Acosta-Piantini et al. [14] showed that hydrolysed sugarcane molasses could serve as an economical substrate for mass production of Lacticaseibacillus paracasei”

2. Minor grammatical errors occur, such as article misuse (“a medium” vs. “the medium”).

R: All grammatical inconsistencies and article misuses have been corrected throughout the manuscript to improve precision and readability.

3. The phrase “black soldier fly larvae cake” is repeated very frequently consider abbreviation (BSFLC) after first mention.

R: We thank the reviewer for this constructive suggestion. The term “black soldier fly larvae cake” has been abbreviated as “BC” after its first mention throughout the text, tables, and figure captions. The choice of BC instead of BSFLC was to comply with the already existing abbreviations such as BCH in the text, tables, figures and their captions and footnotes.

4. Why does the author only consider growth measurements by plate counts (CFU/mL)? Inclusion of growth curves (OD600 vs. time), pH reduction, or lactic acid concentration would improve medium assessment.

R: Thank to the reviewer for this insightful observation. The purpose of growth measurements by plate count was to have an idea of viable biomass, as the strains are targeted to be used for their industrial or probiotic applications and must be alive. Authors like Valle Vargas et al. (2024) cited in our work have also considered measuring viable count at the initial and end of incubation time as a good way to express biomass production in the formulated and control media, Nevertheless, we agree with the reviewer that evaluating other kinetic parameters over time could provide specific information on lag phase, log phase, generation time, etc… But, given the density of results at this phase, we taught to evaluate these kinetic parameters in the next phase that will focus on optimising the strains growth conditions in the compromise medium.

5. The amino acid profile of the by-products (Table 5) is supportive but not central. It could be summarized.

R: We thank the reviewer for this suggestion that will reduce the length of the manuscript. The full amino acid profile has been moved to Supplementary Information (S1 Table), and a short summary remains in the Results section:

See Line 255_Revised manuscript unmarked: “The amino acid profile of the three by-products (S1 Table) shows that……..”

6. Table 8 – Validation of Mathematical Models for Biomass Production: These details could be summarized.

R: As suggested by the reviewer, the detailed model validation parameters (R², AMDA, Bf) for biomass and bacteriocin productions have been transferred to Supporting information (S2 and S3 Tables. The Results section now includes a concise statement. See Lines 367-368 and 505-507_Revised manuscript unmarked:

Reviewer #2

1. Describe more the characteristics or importance of the 3 different sources of the culture medium. In abbreviations, the name or meaning is placed first, and then the corresponding abbreviation in parentheses, and you are interchanged in line 44

R: Thanks for this constructive suggestion. Statements with interchanged abbreviations were canceled due to another reviewer suggestion. Elsewhere, we expanded the introduction to include a clearer description of each by-product.

See Lines 75-85_Revised manuscript unmarked: “The black soldier fly larvae (Hermetia illucens) cake, a protein-rich residue from the biofuel and animal feed industries, represents an emerging biotechnological resource in Africa. It contains high-quality proteins, lipids, minerals (Fe, Zn, Mg), and bioactive compounds which can stimulate microbial growth and metabolism [15]. Pineapple peels, abundantly generated by fruit juice industries and local markets, are rich in fermentable sugars (glucose, fructose, sucrose), organic acids, and B-group vitamins, as well as phenolic compounds that may enhance bacterial stress resistance [16]. Sugarcane molasses, a by-product of sugar refineries such as SOSUCAM in Cameroon, provides readily assimilable carbohydrates, minerals (Ca, K, Mg), and trace elements essential for enzymatic functions in LAB [17]. Their combination may allow the development of economical and sustainable culture media in line with the principles of the circular bioeconomy”

2. (Materials and Methods section): Check the names of the strains because they are misspelled, for example, Lactoccocus is Lactococcus, also the term Mc Farland is McFarland, an example is also in line 97, and in the tables, check each one well.

R: Thanks for this recall. All strain names and technical terms have been double checked and corrected, when necessary, throughout the text and tables.

3. (Materials and Methods section): Improve in general, all the titles of the tables so that they are more descriptive and punctual, because some are too long, and also check the titles of the columns, because some have a capital initial and others do not

R: We thank the reviewer for this helpful suggestion. All table titles have been revised for conciseness and clarity throughout the manuscript. Column headings were standardized, and consistent capitalization was applied throughout.

See Lines 95; 167; 173; 253; 266; 304; 320; 436; 459; 474; 547; 555_Revised manuscript unmarked:

4. (Materials and Methods section): In Table 1 the term status is the function as described in the title, and the case of technological is the application of the strain

R: In Table 1, the term applications has been added to the column heading, as well as in the title.

See Lines 95_Revised manuscript unmarked:

5. (Materials and Methods section):Try to simplify the methodologies for example autoclaved (121°C, 15psi), in the sequence of the pineapple peel hydrolysate, the wording is confusing, there is a lack of volume of water that is used, there is a lack of filters, check also in the black soldier fly larvae cake the equipment is missing team names, volumes and in the solutions the name of the reagent is placed first and then the concentration

Thanks for this valuable suggestion. These concerns were properly addressed. Additionnal equipment, sterilisation pressure and water volume were added. Some sentences were also rephrased.

See Lines 118-176_Revised manuscript unmarked:

6. (Materials and Methods section): Check the number of subtitles. The black soldier fly larvae cake is reserved for later use. It is not specified how they keep it, and they do not mention anything from the number 3 source. Try to be more orderly in writing, follow a sequence in terms of the components of the medium throughout the article try to handle that sequence to describe them

R: We thank the reviewer for this helpful suggestion. The subtitles/subheadings have been fixed to maximum 3 levels. We also improve the readability and flow of this section as well as we maintained the same sequence of the medium component’s terms throughout the manuscript.

See Materials and Methods, Results and Discussion sections_Revised manuscript unmarked:

7. (Materials and Methods section): In line 112, reference is made to the IUPAC and AOAC methods for determining the nutritional profile of the sample, but it is poorly worded to mention what parameters were measured and with what technique, respectively, because it remains as an idea in the air. Because next comes the quantification of minerals, but it is by another type of analysis, according to the wording

R: Thanks for bringing this important point to our attention. This statement has been rewritten while providing which parameter was measured with what techniques

See Lines 118-120_Revised manuscript unmarked: “The physico-chemical analysis of pineapple peels, BC and sugarcane molasses was conducted according to IUPAC methods [20] for fats and sugars, AOAC methods [21] for proteins and ash.”

7. (Materials and Methods section): Line 125 goes first with the name of the acid, followed by the concentration

R: This has been corrected accordingly.

See Line 133_Revised manuscript unmarked: “Samples were first hydrolysed using hydrochloric acid (HCl, 6N)….”

8. (Materials and Methods section): Table 2: Homogenize the initial in all column titles

R: This has been corrected in Table 2 and all the other Tables of the Manuscript.

See Lines 95; 167; 173; 253; 266; 304; 320; 436; 459; 474; 547; 555_Revised manuscript unmarked:

9. (Materials and Methods section): Improve the wording of the title of table 3, it is very long, you can use a table note or better describe it in the text

R: The wording of Table 3 has been improved, as well as that of the other Tables as per the suggestions of the other Reviewer. See Line 173_Revised manuscript unmarked:

10. (Materials and Methods section): Line 180 error in reagent name

R: This has been corrected as NaOH (6N). See Line 158_Revised manuscript unmarked:

11. (Materials and Methods section): In design, it is a mixture design or factorial design

R: It is a mixture design as stated at section Materials and method.

12. (Results section): It is recommended to divide the information into more concise sentences, grouping results by category, for example, maximum values compared to the control. In the text, use different words for significantly, use variants such as notably superior, markedly increased, etc., or place it in the tables of results or methods section.

R: We thank the reviewer for this valuable suggestion. The Results section has been reorganized for better readability and flow. See Lines 242-566_Revised manuscript unmarked:

13. (Results section): It is suggested to create a table with the strain, range (CFU/ml), max vs control, p-value, and write a paragraph that highlights only the key findings to make comparisons clearer in the text

R: We thank the reviewer for this constructive suggestion. New summary tables (Tables 7 and 10) were added, presenting CFU/mL ranges, maximum growth values, and statistical comparisons (p-values) between experimental and the control (MRS/M17) media.

See Lines 320 and 474_Revised manuscript unmarked:

14. (Results section): Check again in this section the names of the strains and units

R: We thank the reviewer for these valuable observations. All strain names have been carefully double checked and corrected, when necessary, throughout the manuscript according to the official taxonomic nomenclature. Measurement units (log CFU/mL, mm, %, etc….) have also been standardised across tables and text to ensure consistency and clarity.

15. (Results section): Explain a little more what the coefficients mean; they only mention linear effects and interactions. This can help in the interpretation to see if there is inhibition or stimulation

R: Thanks for this observation. We have expanded the interpretation in the Discussion as:

“For all the strains, the SM (x3), the interaction of BCH and PPH (x1x2), as well as the interaction of PPH and SM (x2x3) showed positive effects on biomass production, except for Lacticaseibacillus rhamnosus 347. In addition, the lin

---

## [Editor Report · Decision Letter 1]

12 Nov 2025

Development of a low-cost culture medium from industrial and environmental by-products for sustainable cultivation of Lactic Acid Bacteria

PONE-D-25-35478R1

Dear Dr. Pierre Marie,

We’re pleased to inform you that your manuscript has been judged scientifically suitable for publication and will be formally accepted for publication once it meets all outstanding technical requirements.

Kind regards,

Guadalupe Virginia Nevárez-Moorillón, Ph.D.

Academic Editor

PLOS ONE

Additional Editor Comments (optional):

Please revise that the final document is the last version, correponding to the highlighted version.
---

## [Editor Report · Acceptance letter]

PONE-D-25-35478R1

PLOS ONE

Dear Dr. KAKTCHAM ,

I'm pleased to inform you that your manuscript has been deemed suitable for publication in PLOS ONE. Congratulations! Your manuscript is now being handed over to our production team.

Kind regards,

on behalf of

Dr. Guadalupe Virginia Nevárez-Moorillón

Academic Editor

PLOS ONE